# Mechanochemistry of phosphate esters confined between sliding iron surfaces

Carlos Ayestarán Latorre[1,2], Joseph E. Remias[3], Joshua D. Moore[3,6], Hugh A. Spikes[1], Daniele Dini [1,4,5] & James P. Ewen [1,4,5]✉

The molecular structure of lubricant additives controls not only their adsorption and dissociation behaviour at the nanoscale, but also their ability to reduce friction and wear at the macroscale. Here, we show using nonequilibrium molecular dynamics simulations with a reactive force field that tri(s-butyl)phosphate dissociates much faster than tri(n-butyl) phosphate when heated and compressed between sliding iron surfaces. For both molecules, dissociative chemisorption proceeds through cleavage of carbon—oxygen bonds. The dissociation rate increases exponentially with temperature and stress. When the rate—temperature—stress data are fitted with the Bell model, both molecules have similar activation energies and activation volumes and the higher reactivity of tri(s-butyl)phosphate is due to a larger pre-exponential factor. These observations are consistent with experiments using the antiwear additive zinc dialkyldithiophosphate. This study represents a crucial step towards the virtual screening of lubricant additives with different substituents to optimise tribological performance.

[1] Department of Mechanical Engineering, Imperial College London, South Kensington Campus, London SW7 2AZ, UK. [2] Department of Materials, Imperial College London, South Kensington Campus, London SW7 2AZ, UK. [3] Afton Chemical Corporation, Richmond, VA 23219, USA. [4] Institute of Molecular Science and Engineering, Imperial College London, South Kensington Campus, London SW7 2AZ, UK. [5] Thomas Young Centre for the Theory and Simulation of Materials, Imperial College London, South Kensington Campus, London SW7 2AZ, UK. [6] Present address: Dassault Systèmes Americas Corporation, Waltham, MA 02451, USA. ✉email: j.ewen@imperial.ac.uk

Mechanochemistry describes a diverse range of phenomena where chemical reactivity is influenced by mechanical load[1]. It could soon play a central role in many important industrial processes[1], such as ammonia production[2]. Currently, the most economically valuable application of mechanochemistry is for the formation of friction- and/or wear-reducing films by lubricant additives on rubbing surfaces[3]. These 'tribofilms' are critical to maintaining the efficient and reliable operation of the majority of lubricated machine components. Zinc dialkyldithiophosphate (ZDDP) is the most popular antiwear additive; it is included in almost all engine oils, as well as many other types of lubricant[4]. It forms relatively thick (~100 nm), patchy tribofilms on rubbing steel surfaces, which are mostly composed of zinc and iron polyphosphates. Due to environmental concerns with ZDDP, there has been a sustained global research effort over several decades to develop alternatives that are metal-free, form lower amounts of sulfated ash when combusted, and contain less phosphorus and sulfur[5]. In macroscale tribometer experiments, metal-free dialkyldithiophosphates (DDPs)[6], sulfur-free zinc dialkylphosphates (ZDPs)[7] and metal- and sulfur-free trialkylphosphates[8] have all been shown to form protective tribofilms on steel surfaces; however, these are generally thinner and grow more slowly compared to those formed by ZDDP. Consequently, low-sulfated ash, phosphorus, and sulfur (SAPS) additives generally show poorer antiwear performance than ZDDP[5].

An improved understanding of the atomic-scale behaviour of antiwear additives inside tribological contacts is required to rationally design new molecules with improved performance and reduced environmental impact[9]. This information is challenging to obtain from macroscale tribometer experiments in the mixed/boundary lubrication regime[10]. In these experiments, numerous solid−solid asperity contacts occur, resulting in local conditions that are difficult to measure and control. In recent years, in situ experiments of single nanoscale asperities have enabled the study of additive behaviour under more controlled conditions[11]. For example, Gosvami et al.[12] showed using atomic force microscopy (AFM) that the tribofilm formation rate of ZDDP is exponentially dependent on both temperature, $T$, and normal stress, $\sigma_{zz}$. These observations are consistent with a mechanochemical driving force for tribofilm formation[12]. Tribofilm growth in these experiments can be modelled as a stress-augmented thermally activated (SATA) process[13]. The rate constant for SATA processes can be calculated through a modified Arrhenius equation,

$$k = A \exp\left(-\frac{E_a - \sigma \Delta V^*}{k_B T}\right) \qquad (1)$$

where $A$ is the pre-exponential factor, $E_a$ is the activation energy, $\sigma$ is the applied stress, $\Delta V^*$ is the activation volume, and $k_B$ is the Boltzmann constant. This equation was first proposed in the context of molecular systems by Evans and Polanyi[14], but it is now usually known as the Bell model[15]. Experiments at different scales have suggested that ZDDP tribofilm formation follows zero-order (rate = $k$)[16] or fractional-order (rate = $k[ZDDP]^n$)[12] kinetics. These observations suggest that the ZDDP molecules adsorb onto the steel surface before they dissociate and form tribofilms[16]. Similar mechanochemical responses have also been observed for metal-free DDPs in AFM experiments by Dorgham et al.[17], although these additives produced thinner tribofilms that formed more slowly than with ZDDP.

Zhang and Spikes[18] showed that under full-film elastohydrodynamic lubrication (EHL) conditions, ZDDP tribofilms formed on tungsten carbide surfaces from high shear stress traction fluids, but not from low shear stress lubricant base oils[18]. Using Eq. (1), they showed that the rate of tribofilm formation

had an exponential dependence on the shear stress ($\sigma_{xy}$), rather than the normal stress ($\sigma_{zz}$)[16,18]. Zhang et al.[16] recently confirmed this finding using steel surfaces and ZDDPs containing different alkyl substituents. In similar tribometer experiments, Ueda et al.[19] showed that tribofilm thickness was the largest for steel substrates, followed by silicon nitride and then tungsten carbide, with no tribofilm formation observed on silicon carbide or diamond-like-carbon (DLC). Along with accompanying quartz crystal microbalance (QCM) experiments, these observations suggested that strong surface adsorption is required to form a tribofilm and that the rate of tribofilm formation was faster on harder surfaces, which led to higher contact stresses[19].

Antiwear additives with different alkyl and aryl substituents can show very different thermal, mechanochemical and tribochemical reactivity. Dickert and Rowe[20] suggested that secondary dialkyl ZDDPs have lower thermal stability compared to primary dialkyl ZDDPs because the former are more susceptible to $\beta-H$ elimination reactions. Jones and Coy[21] showed that the thermal stability of ZDDPs increased in the order; secondary alkyl < primary linear alkyl < primary branched alkyl < aryl. They suggested that primary and secondary ZDDPs undergo different decomposition mechanisms; alkyl transfer and $\beta-H$ elimination, respectively[21]. Fujita et al.[10] noted that secondary ZDDPs formed thicker tribofilms more quickly than primary ZDDPs on steel substrates under mixed/boundary lubrication (tribochemical) conditions. Recently, Zhang et al.[16] showed that ZDDPs with secondary dialkyl groups also form tribofilms much faster than those with primary dialkyl groups on steel surfaces under full-film EHL (mechanochemical) conditions. Similar substituent effects have also been observed for other antiwear additives, such as ZDPs and phosphate esters. For example, triarylphosphates are known to be more thermally stable than trialkylphosphates[22]. Similar to ZDDP[20], this could be due to the susceptibility of trialkylphosphates to $\beta-H$ elimination reactions, which cannot occur for triarylphosphates[23,24]. Moreover, Hoshino et al.[7] showed that secondary ZDPs form tribofilms more quickly than primary ZDPs on rubbing steel surfaces. These observations imply that the rate-determining step for tribofilm growth by ZDDP, ZDP, and phosphate ester antiwear additives on rubbing steel surfaces is the initial removal of the alkyl groups. This is perhaps unsurprising given that dissociative chemisorption is often the rate-determining step for heterogeneously-catalysed processes[25]. To design new antiwear additives with controlled tribofilm formation rates, it is therefore critical to obtain a detailed understanding of the effects of different alkyl and aryl substituents on mechanochemical reactivity.

It has recently been demonstrated that molecular dynamics (MD) simulations can be used to virtually screen and even autonomously design new lubricant molecules with a high viscosity index[26]. MD simulations have also been successfully applied to compare the thermal stability of different antiwear additives on steel surfaces. In particular, Ewen et al.[27] have studied substituent effects on the thermal decomposition of phosphate esters on ferrous surfaces, reproducing the same order of reactivity as observed experimentally for ZDDP[21]. Moreover, nonequilibrium molecular dynamics (NEMD) simulations have provided unique insights into mechanochemical processes at molecule-solid interfaces inside tribological systems[28]. For example, NEMD simulations have recently been used to study the tribopolymerization of aldehydes between alumina surfaces[29], phosphoric acid[30,31], allyl alcohols[32] and terpenes[33] between sliding silica surfaces, and cyclopropane carboxylic acid between iron oxide surfaces[34]. They have also been used to study the mechanochemical decomposition of alkyl sulfides between sliding iron surfaces[35] and perfluoropolyethers (PFPEs) between DLC surfaces[36], the vapour phase lubrication of trialkylphosphates[37]

and trialkylphosphites[38,39] between iron surfaces, and the ultra-low friction of organic friction modifier additives between DLC surfaces[40]. Most of these NEMD simulations employed many-body empirical force fields[30–36], while others use first principles methods[29,37–39], or sometimes a combination of both of these techniques[40].

First principles methods are extremely computationally expensive and are therefore limited to single molecules, sub-nanosecond timescales, and very severe conditions[29,37–40]. Bond order potentials are several orders of magnitude cheaper than first principles approaches[41], allowing much larger systems to be simulated under experimentally-relevant conditions. In particular, ReaxFF, which was originally developed by van Duin et al. to study hydrocarbon reactivity[42], has been parameterised to model a wide range of chemical systems and processes[43]. Of particular relevance to this study, ReaxFF simulations are now routinely used to study tribochemical reactions[44]. The high parallel efficiency (linear-scaling) of ReaxFF MD simulations allows the routine study of large systems (several thousands of atoms[43]) under conditions that are much closer to, or even the same as, those used experimentally[41]. For some systems, it has been argued that current ReaxFF parameterisations are incapable of accurately reproducing chemical reactions that are observed under tribological conditions using first principles methods[45]. However, for many important chemical processes (e.g. heterogeneous catalysis, atomic layer deposition, and nanoindentation), careful parametrisation has ensured that the accuracy of ReaxFF MD simulations is close to that obtained when using first principles methods[43]. This includes the adsorption and dissociation of phosphate esters on iron and iron oxide surfaces[46].

In this study, we use NEMD simulations with ReaxFF to study the reactivity of trialkylphosphate esters heated and compressed between sliding iron surfaces. We compare the decomposition of tri(n-butyl)phosphate (TNBP), which contains primary linear alkyl substituents, compared to secondary tri(s-butyl)phosphate (TSBP), which contains secondary alkyl groups (Fig. 1a). The trends between the kinetic parameters obtained from our NEMD simulations for primary and secondary trialkylphosphates (Fig. 1b) agree well with those obtained from recent macroscale tribometer experiments using primary and secondary ZDDPs[16]. This study gives new insights into the rate-determining step for tribofilm formation (dissociative chemisorption) and its dependence on temperature and stress. It also represents an important step towards the virtual screening and autonomous molecular design[26] of antiwear additives with optimised molecular structures for tailored mechanochemical and tribological responses.

## Results and discussion
**Stress conditions**. First, we studied the variation in the shear stress, $\sigma_{xy}$, with normal stress, $\sigma_{zz}$, for the different systems and conditions studied. This is important since, in previous experiments, the shear stress, rather than the normal stress, was shown to control the mechanochemical reactivity of antiwear additives[16,18]. Figure 1c and d show how the mean shear stress varies with the mean normal stress during steady-state sliding for TNBP and TSBP respectively. The shear stress increases linearly with the normal stress with a finite intercept. Therefore, the friction coefficient, $\mu$, values shown in Fig. 1c and d were calculated using the version of the Amontons–Coulomb friction equation extended by Derjaguin[47]. Here, $\sigma_{xy} = \mu \, \sigma_{zz} + \sigma_0$, where $\sigma_0$ is the Derjaguin offset, which is related to adhesion at the sliding interface. The mean shear stress is identical for TNBP and TSBP within statistical uncertainty. In general, the mean shear stress is slightly higher at a lower temperature, although the friction coefficient remains essentially unchanged for the different

systems and temperatures considered. The steady-state shear stress values shown in Fig. 1 were used in the Bell model[15] (Eq. 1) to calculate the activation energy, activation volume, and pre-exponential factor of TNBP and TSBP dissociation[13].

**Dissociation mechanism**. Figure 2 shows the change in the number of covalent bonds in the trialkylphosphate molecules with sliding time. The first 0.1 ns is the equilibration phase at 300 K and 10 MPa with no sliding, while the remaining 1.0 ns is the heating ($T = 400$ K), compression ($\sigma_{zz} = 2$ GPa), and sliding ($v_s = 10$ m s$^{-1}$) phase. Figure 2 shows that the initial dissociation reaction is C−O cleavage for both TNBP and TSBP. This observation is consistent with recent MD simulations of the thermal decomposition of TNBP and TSBP on α-Fe(110) and Fe$_3$O$_4$(001) surfaces[27]. The rate of C−O cleavage is faster for TSBP than TNBP, which is due to the greater stability of secondary alkyl cations or radicals compared to primary ones[49]. Note that heterolysis and homolysis cannot be differentiated using the standard ReaxFF approach because the electrons are not explicitly modelled. Previous experimental studies have suggested that heterolytic C−O fission is probably the dominant process during the thermal decomposition of trialkylphosphates[50]. For TSBP, some C−O cleavage reactions occur even during equilibration at relatively low temperature (300 K) and pressure (10 MPa). The higher reactivity of TSBP than TNBP can be rationalised through the inductive effect, because alkyl groups have more electron-donating character compared to hydrogen atoms[51]. There is also evidence in the literature suggesting that steric effects and hyperconjugation can also affect the susceptibility of phosphate esters to C−O bond cleavage;[49] however, the substituent effect sobserved in the current simulations suggest that such effects are less important for these systems. In the current NEMD simulations, C−O cleavage through dissociative chemisorption is favoured over $\beta$−H elimination, which has previously been suggested to be the major route of C−O cleavage for both trialkylphosphates[23,24] and ZDDPs[20,21]. Dissociative chemisorption is promoted by the stabilisation of the resultant ions or radicals formed by the iron surface[38,39]. Some of the broken C−O bonds reform for TNBP (blue dotted lines), whereas this is negligible for TSBP.

The pyrolysis of TNBP is known to proceed mostly through C−O cleavage[50]. However, there is some disagreement in the experimental literature regarding the thermal decomposition mechanism of trialkylphosphates on iron oxide surfaces. Some studies using TNBP have suggested that this proceeds through C−O cleavage[23,24], while others have postulated that P−O cleavage dominates[52]. The latter suggestion was based upon the detection of gas-phase alcohols, rather than alkenes when TNBP was gradually heated to 800 K on an iron oxide surface[52]. However, recent MD simulations of TNBP showed that gas-phase alcohols could also be formed by oxidation of the surface-adsorbed alkyl groups (formed by C−O cleavage) by the iron oxide surface, followed by desorption[27]. The large substituent effects observed in previous MD simulations[27] can only be accounted for if C−O cleavage, rather than P−O cleavage, is the rate-determining step for the thermal decomposition of trialkylphosphates on ferrous surfaces. More specifically, phosphate esters containing secondary alkyl groups (e.g. TSBP), that form more stable carbocations or radicals by C−O cleavage[49], decomposed much faster than those with primary groups (e.g. TNBP)[27]. The current NEMD simulations show that the same dissociation mechanism dominates under mechanochemical conditions.

The polyphosphate tribofilms formed from trialkylphosphates and ZDDP grow through nucleophilic substitution reactions. After C−O cleavage has occurred, the resultant

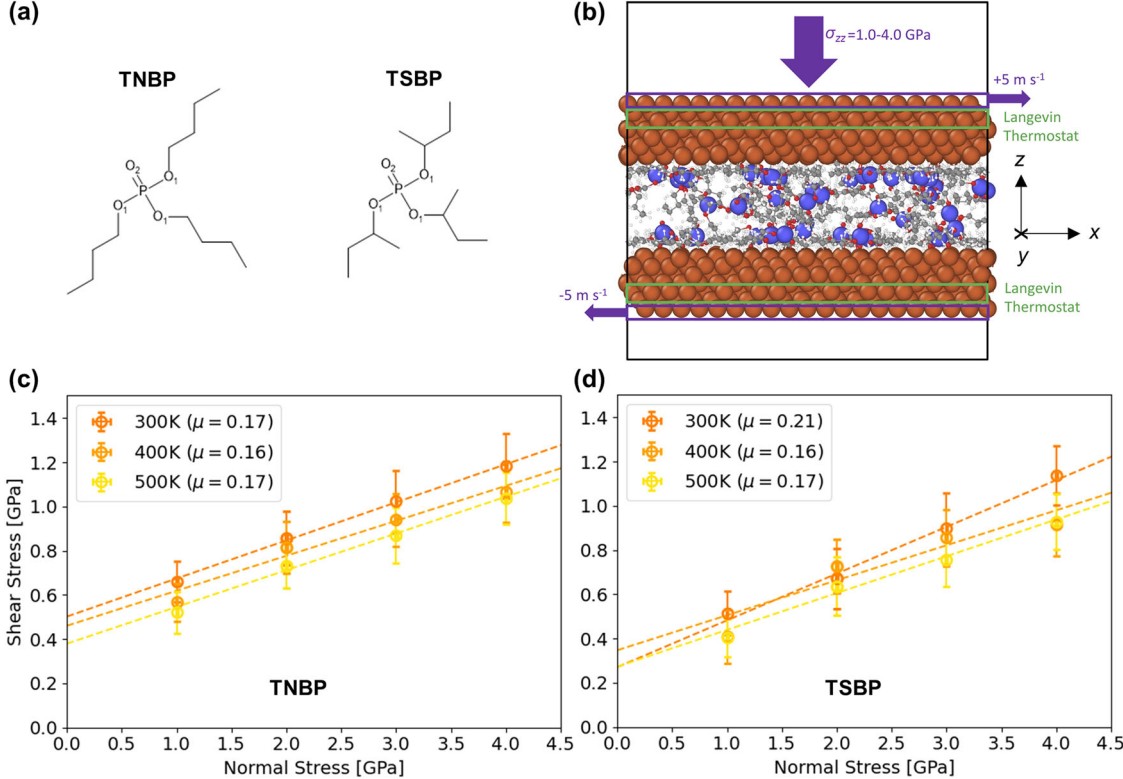

**Fig. 1 Systems studied and stress conditions.** Molecular structures of phosphate esters considered: TNBP and TSBP (**a**). Oxygen atoms in P−O bonds are labelled $O_1$, while those in P=O bonds are labelled $O_2$. Snapshot showing a representative system of TNBP molecules confined between α-Fe(110) substrates after equilibration (**b**). Rendered with OVITO[48], Fe atoms are shown in orange, O in red, P in blue, C in black, and H in white. Variation in mean shear stress with normal stress for TNBP (**c**) and TSBP (**d**) between 300−600 K during steady-state sliding. Dashed lines are fits to Amontons–Coulomb friction equation extended by Derjaguin[47]. The friction coefficient, $\mu$, for each case is shown in the legend in parenthesis (±0.05). Vertical bars represent one standard deviation between the block-averaged (0.1 ns) shear stress values, horizontal bars for the normal stress are smaller than the symbol size.

dialkylphosphate or monoalkylphosphate anion or radical is more nucleophilic than the corresponding trialkylphosphate. Moreover, the central P(V) atom is susceptible to nucleophilic attack by neighbouring molecules due to reduced steric hindrance[21]. Figure 2a, b show that P−O bonds begin to break soon after the C−O bonds. Concomitantly with this process, new P−O bonds between the phosphate ester molecules are formed, confirming the nucleophilic substitution mechanism (Fig. 2c). The rate of P−O bond formation is faster than the rate of P−O bond cleavage, meaning there is an overall increase in P−O bonding. This indicates that polyphosphate chains are beginning to grow as P−O−R bonds are replaced by P−O−P bonds[21]. Inside lubricated contacts, these nucleophilic substitution reactions would eventually result in the formation of thick (~100 nm) polyphosphate tribofilms on the rubbing steel surfaces. This process has recently been observed experimentally for systems lubricated by a secondary tri(*i*-propyl)phosphate dissolved in a hydrocarbon base oil[8]. Comparing the green dotted lines in Fig. 2a, b, the rates of P−O cleavage and formation are both faster for TSBP than TNBP. As for the initial C−O cleavage, the subsequent nucleophilic substitution reaction can also be mechanochemically-promoted[53]. However, since nucleophilic substitution is not the rate-determining step, we restrict our detailed rate analysis to the C−O cleavage process.

We also performed a subset of MD simulations under identical conditions to those shown in Fig. 2 (2 GPa and 400 K) without sliding. As shown in Supplementary Fig. 1, the decomposition mechanism is the same as under sliding and TSBP is still more reactive than TNBP; however, reactivity for both molecules is much lower than under sliding conditions (Fig. 2a, b). After the

initial compression phase (0.1 ns), there is virtually no further C−O cleavage or nucleophilic substitution. This finding supports the experimental observation for ZDDP that the shear stress, rather than the normal stress, drives the mechanochemical decomposition process[16,18].

**Dissociation rate.** The dissociation rate of TNBP and TSBP molecules between the sliding α-Fe(110) surfaces was investigated over a wide range of temperature and stress conditions. Figure 3a (TNBP) and Fig. 3b (TSBP) show how the number of intact phosphate ester molecules decays with sliding time at a fixed temperature ($T = 400$ K) and variable pressure ($\sigma_{zz} = 1–4$ GPa). Fig. 3c (TNBP) and Fig. 3d (TSBP) show the same relationship at a fixed pressure ($\sigma_{zz} = 2$ GPa) and variable temperature ($T = 300−500$ K). The results for some of the other conditions considered are shown in Supplementary Fig. 2. A small fraction (0−20%) of the TNBP and TSBP molecules dissociate during the equilibration phase, which are not included in the fitting process used to obtain the rates. During the heating, compression, and sliding phase, there is an exponential decay in the number of intact TNBP and TSBP molecules with simulation time, which is indicative of a first-order reaction[36]. The NEMD simulation data (dashed lines) during the production phase are therefore fitted with an exponential function (solid lines) to determine the reaction rate. In previous experiments for ZDDP[12,16,17] and DDP[17], the rate of tribofilm formation has followed either zero-order or fractional-order kinetics. The main reason for this difference is the finite number of additive molecules in the NEMD simulations, which means that the dissociation rate inevitably

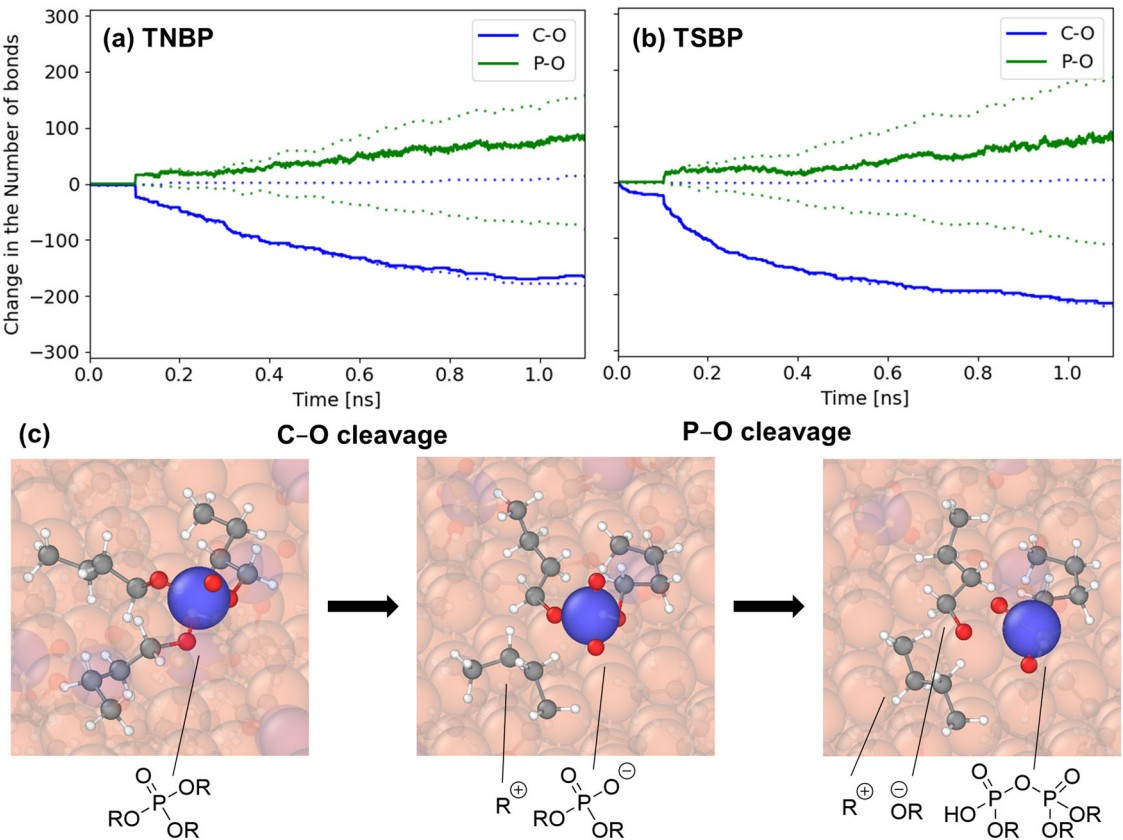

**Fig. 2 Decomposition mechanism of phosphate esters.** Change in the number of covalent bonds with sliding time for TNBP (**a**) and TSBP (**b**). Representative examples shown at $v_s = 10\,\mathrm{m\,s^{-1}}$, $T = 400\,\mathrm{K}$ and $\sigma_{zz} = 2\,\mathrm{GPa}$. Dashed lines show the individual contributions from the bond formation (positive) and bond cleavage (negative), while solid lines show the change in the total number of bonds. Snapshots showing C−O cleavage (dissociative chemisorption) and P−O cleavage (nucleophilic substitution) for a single TNBP molecule during one of the NEMD simulations (**c**). Chemical species are labelled assuming that the C−O bonds are cleaved through heterolysis[50]. Surface and molecules where bonds are not broken are translucent. Fe atoms are shown in orange, O in red, P in blue, C in black and H in white, rendered with OVITO[48].

decreases as fewer molecules are available to react. In the experiments, the additive molecules are continually replenished from the base oil solution and the bulk concentration does not significantly decrease[16]. The rate of tribofilm formation does not change appreciably over the course of the experiments[16], meaning that the rate-determining step is zero-order and involves surface-adsorbed molecules[54].

At constant temperature, the decomposition rate of TNBP (Fig. 3a) and TSBP (Fig. 3b) both increase with increasing pressure. As previously noted from previous MD simulations of thermal decomposition[27], the dissociation rate is much higher for TSBP (Fig. 3b) than TNBP (Fig. 3a). At constant pressure, the dissociation rate of TNBP (Fig. 3c) and TSBP (Fig. 3d) both increase with increasing temperature. The dissociation rate is again much higher for TSBP (Fig. 3d) compared to TNBP (Fig. 3c) for all the conditions studied.

**Mechanochemical parameters**. Figure 4a, b show the shear stress dependence of the dissociation rates at different temperatures for TNBP and TSBP, respectively. The shear stress values were taken from Fig. 1, while the dissociation rates were calculated from the exponential decay curves in Fig. 3 and Supplementary Fig. 2. For both TNBP and TSBP, the reaction rate increases exponentially with shear stress, which is indicative of a SATA process[13]. The activation volume, $\Delta V^*$, was calculated from two-dimensional (2D) fits the data using Eq. (1) from $T = 300-500\,\mathrm{K}$. Using this approach, $\Delta V^*$ increases with increasing temperature from $15 \pm 6\,\mathrm{\AA^3}$ at 300 K to $29 \pm 4\,\mathrm{\AA^3}$ at 500 K for TNBP and from

$8 \pm 3\,\mathrm{\AA^3}$ at 300 K to $28 \pm 24\,\mathrm{\AA^3}$ at 500 K for TSBP (Supplementary Table 1). In previous experimental studies of the mechanochemistry of antiwear additives[12,16,17], $\Delta V^*$ has been treated as a reaction constant, which is not temperature-dependent. However, these experiments have typically been performed over a much narrower temperature range than was used in the current NEMD simulations. Previous experiments have shown that $\Delta V^*$ can be temperature-dependent for other processes, such as self-diffusion in zinc metal[55] and dislocation motion in steel[56]. Previous NEMD simulations with ReaxFF have shown that $\Delta V^*$ can also be pressure-dependent for the decomposition of PFPEs between DLC surfaces[36]. The increase in $\Delta V^*$ with increasing temperature observed in the current simulations could be due to a reduction in the contact stiffness. A recent DFT study of hydroxylated silica−silica interfaces has shown that $\Delta V^*$ is inversely proportional to the contact stiffness[57]. It has been shown experimentally that the elastic modulus of bulk α-Fe decreases by around 10% as the temperature is increased between 300−500 K[58]. We observe a similar reduction in elastic modulus over this temperature range for our thin α-Fe slabs. Therefore, the large increase (>100%) in $\Delta V^*$ we observe cannot be completely accounted for by the change in contact stiffness of the solid surfaces. As expected for SATA processes[13], the dissociation rate also increases exponentially with temperature, as shown in the Arrhenius plots in Supplementary Fig. 3.

The activation energy, $E_a$, calculated from the 2D fits to Eq. (1), increases roughly linearly with increasing pressure from $11 \pm 4$ to $18 \pm 4\,\mathrm{kJ\,mol^{-1}}$ for TBNP and $11 \pm 7$ to $22 \pm 10\,\mathrm{kJ\,mol^{-1}}$ for TSBP between 1−4 GPa (Supplementary Table 2). The natural

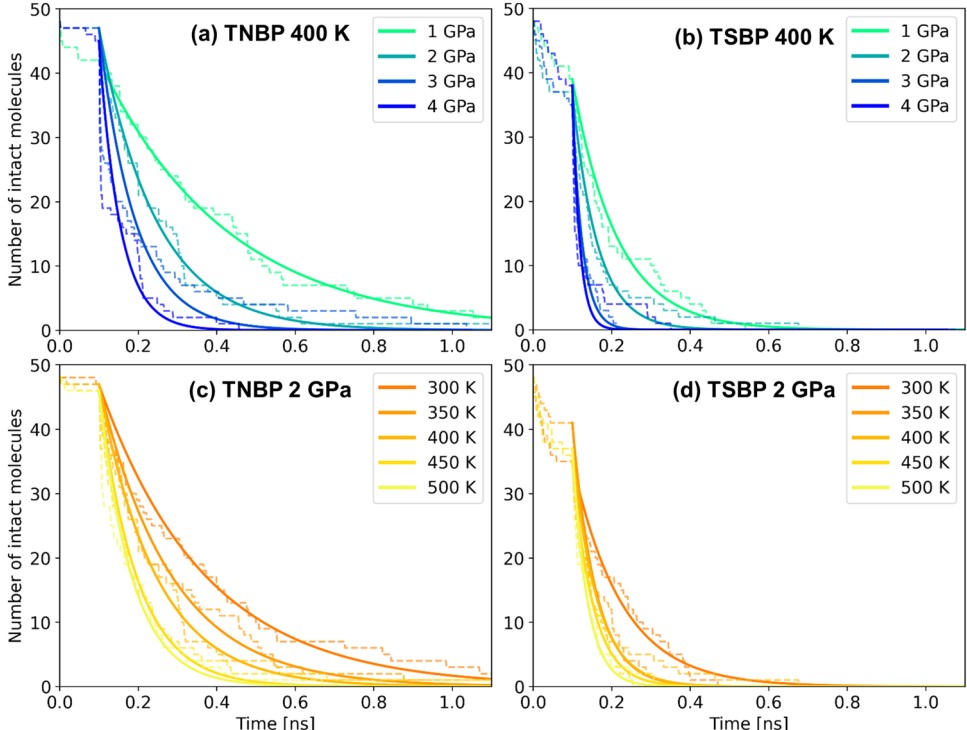

**Fig. 3 Dissociation rate of phosphate esters under different conditions.** The effect of pressure on the number of intact phosphate ester molecules with sliding time for TNBP (**a**) and TBSP (**b**) at constant temperature ($T = 400$ K). The influence of temperature on the number of intact phosphate ester molecules with sliding time for TNBP (**c**) and TBSP (**d**) at constant pressure ($\sigma_{zz} = 2$ GPa). Dashed lines are results from the NEMD simulations; solid lines are fits to the data assuming exponential decay.

logarithm of the pre-exponential factor, $\ln(A)$, also increases with increasing pressure from $24 \pm 1$ to $26 \pm 4$ for TNBP and $25 \pm 2$ to $28 \pm 2$ for TSBP over the same range (Supplementary Table 2). A concurrent increase in these parameters ($E_a \propto \ln[A]$) is known as the kinetic compensation effect (Supplementary Fig. 4). This effect has frequently been observed for heterogeneously-catalysed processes[54]. The fact that a compensation effect is observed here implies that the changes in these values could be mathematical in origin, rather than suggesting any physical changes to the reaction pathway[59].

To understand the combined effects of temperature and stress on reactivity, the decomposition rate data was also fit to Eq. (1) as a 3D surface[17]. Fits to the logarithm of the rate are used to prevent biasing of the 3D fits to the higher rates. The 3D fits are shown for TBNP in Fig. 4c and for TSBP in Fig. 4d. The 3D surfaces were used to calculate $\Delta V^*$, $E_a$, and $A$ values over the entire range of temperatures and pressures studied (Table 1). Unlike the 2D plots, the 3D fitting method does not capture the change in the parameters with temperature and pressure, which we suggest are only physically meaningful for $\Delta V^*$. However, compared to the 2D fits (Supplementary Table 1 and Supplementary Table 2), the 3D fits (Table 1) generally lead to reduced uncertainties in all three parameters in the Bell model[15]. The uncertainties in $E_a$ and $\Delta V^*$ from the 3D fits are smaller than those obtained from 2D fits of Eq. (1) to experimental ZDDP tribofilm growth data obtained using AFM[12]. The uncertainty in $A$ is much smaller than from this previous AFM study, where possible values spanned three orders of magnitude. The 3D fits to Eq. (1) using the parameters in Table 1 accurately describe the rates from the NEMD simulations over the entire range of conditions studied. There was good agreement between the logarithm of the predicted and calculated rates (Supplementary Fig. 7) for both TNBP ($R^2 = 0.89$) and TSBP ($R^2 = 0.82$). The parameters for Eq. (1) obtained from the 3D (Table 1) and 2D fits (Supplementary Tables 1, 2) are in good

agreement. The values of $\Delta V^*$ from the 3D fits are close to those obtained from the 2D fits at intermediate temperature (Supplementary Fig. 5). Similarly, the $E_a$ (Supplementary Fig. 6) and $\ln(A)$ (Supplementary Fig. 7) values from the 3D fits are both close to those obtained from the 2D fits at an intermediate pressure. It is worth noting that the calculation of $E_a$ and $A$ from the 2D Arrhenius plots (Supplementary Fig. 3) and the Bell model[15] requires a fixed value for $\Delta V^*$. For the values reported in Supplementary Table 2, we used the mean value obtained from the 2D rate versus shear stress fits (Fig. 4a, b) from $300-500$ K. On the other hand, for the 3D fits, all three parameters are calculated concurrently using the entire dataset. This facilitated the considerably reduced uncertainty in $\Delta V^*$ (Supplementary Fig. 5) $E_a$ (Supplementary Fig. 6) and $\ln(A)$ (Supplementary Fig. 7) from the 3D fitscompared to the 2D fits.

From the 3D fits, $\Delta V^*$ is the same for TNBP ($16.4 \pm 2.8$ Å$^3$) and TSBP ($16.7 \pm 4.0$ Å$^3$) within the uncertainty of the calculations. From previous macroscale tribometer experiments[16], $\Delta V^*$ was also found to be identical ($15$ Å$^3$) for ZDDPs with primary and secondary substituents. The slightly larger $\Delta V^*$ values calculated for trialkylphosphates compared to ZDDPs suggest that the former are more mechanochemically susceptible. If the normal stress, $\sigma_{zz}$, rather than the shear stress, $\sigma_{xy}$, is used as the variable in Eq. (1), then the activation volumes would decrease by approximately a factor of five ($1/\mu$ in Fig. 1). This approach yields a $\Delta V^*$ value for TNBP and TSBP that is the same order of magnitude as that measured for mixed primary/secondary ZDDPs ($4 \pm 1$ Å$^3$) using AFM[12]. The fact that the $\Delta V^*$ values are similar for primary and secondary trialkylphosphates and ZDDPs suggests that, although the former dissociates and forms tribofilms much more slowly, the differences cannot be attributed to their relative mechanochemical susceptibilities.

The $\Delta V^*$ values can also be compared to the molecular volumes[60], which were calculated for TNBP and TSBP molecules

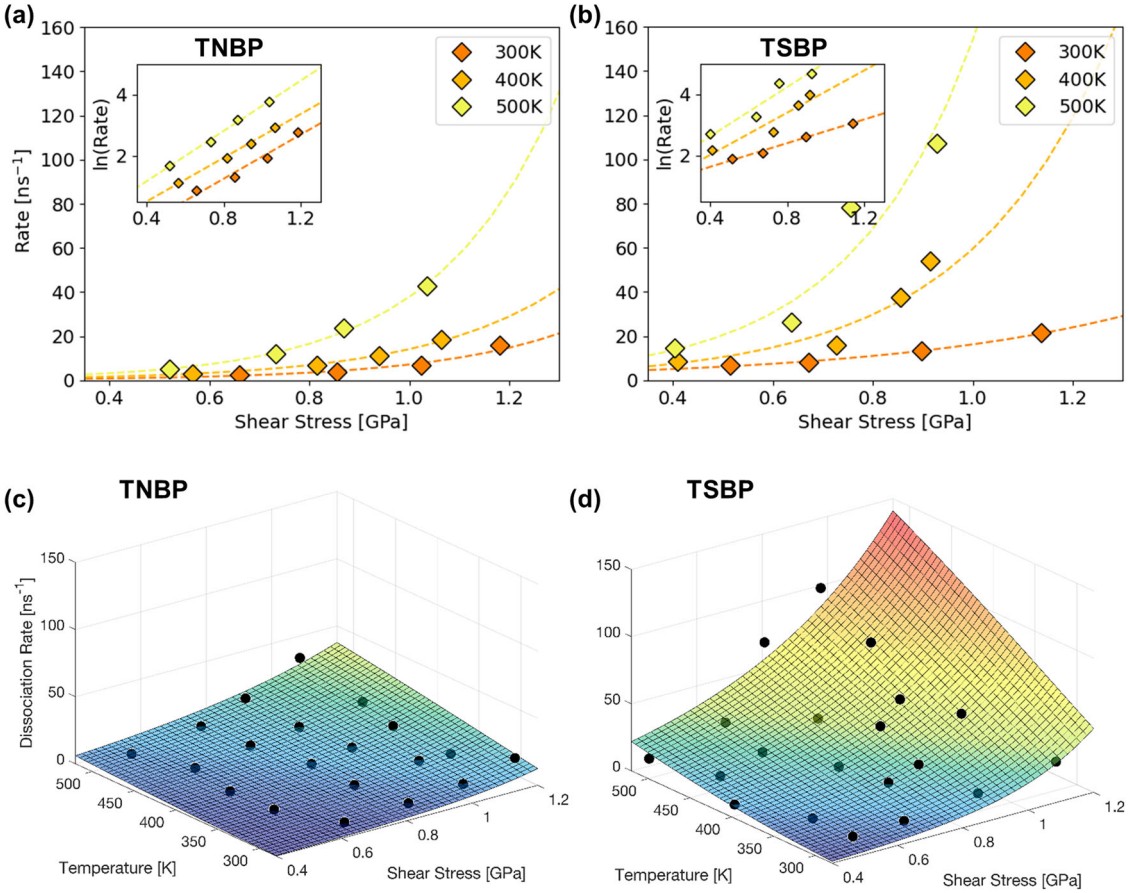

**Fig. 4 2D and 3D fits of the rate data to the Bell Model.** Shear stress dependence of the dissociation rate for TNBP (**a**) and TSBP (**b**) at different temperatures. Insets show the same data on a logarithmic $y$-axis. 3D plots showing the variation in dissociation rate with temperature and shear stress for TNBP (**c**) and TSBP (**d**). 3D surfaces are fits of the logarithm of the rates to the Bell model[15].

**Table 1 Calculated values of the activation energy, $E_a$, natural logarithm of the pre-exponential factor, ln($A$), and activation volume, $\Delta V^*$, for TNBP (Fig. 4c) and TSBP (Fig. 4d) from the 3D fits to Eq. (1). The parameter ranges represent the 95% confidence intervals from the 3D fits.**

|  | $E_a$ (kJ mol$^{-1}$) | ln($A$) (s$^{-1}$) | $\Delta V^*$ (Å$^3$) |
|---|---|---|---|
| TNBP | 17.4 ± 2.6 | 25.6 ± 0.6 | 16.4 ± 2.8 |
| TSBP | 17.5 ± 3.6 | 26.9 ± 0.9 | 16.7 ± 4.0 |

relaxed using the MM2 force field[61]. Using a probe size of 3 Å$^3$, TNBP has a molecular volume, $V \approx 440$ Å$^3$ and for TSBP, $V \approx 390$ Å$^3$. Thus, $\Delta V^*$ represents a ~4% deformation of the molecule for both TNBP and TSBP. This percentage deformation is comparable to that calculated from previous NEMD simulation results for α-pinene (~3%)[33] and allyl alcohol (~7%)[32] oligomerisation between sliding silica surfaces, as well as the decomposition of PFPEs (~2%)[36] between DLC surfaces.

The activation energy, $E_a$, calculated from the 3D fits to Eq. (1) in Fig. 4c, 4d is also very similar for TNBP (17.4 ± 2.6 kJ mol$^{-1}$) and TSBP (17.5 ± 3.6 kJ mol$^{-1}$). The thermal decomposition of TNBP in an inert atmosphere (170 kJ mol$^{-1}$)[50] and with a steel catalyst in the air (80 kJ mol$^{-1}$)[22] yielded higher $E_a$ values, suggesting that the application of stress alters the reaction pathway to considerably reduce the energy barrier[62]. The calculated values of $E_a$ for TNBP and TSBP are within the range observed from AFM experiments using DDP (10−25 kJ mol$^{-1}$)[17]. However, the $E_a$ values are

somewhat lower than those obtained from AFM experiments for mixed primary/secondary ZDDPs (77 ± 19 kJ mol$^{-1}$)[12] and full-film EHL experiments for primary (59 kJ mol$^{-1}$) and secondary (84 kJ mol$^{-1}$) ZDDPs[16]. This implies that there is a lower energy barrier for removal of the alkyl groups in trialkylphophates than ZDDPs under mechanochemical conditions.

The $\Delta V^*$ and $E_a$ values from the 3D fits for TNBP (Fig. 4c) and TSBP (Fig. 4d) are almost identical (Table 1). Thus, the faster mechanochemical dissociation of TSBP than TNBP cannot be rationalised through the calculated $E_a$ and $\Delta V^*$ values. On the other hand, the pre-exponential factor, $A$, is approximately four times greater for TSBP ($1.4 \pm 1.1 \times 10^{11}$ s$^{-1}$) than TNBP ($5.0 \pm 3.0 \times 10^{11}$ s$^{-1}$), as shown in Table 1. The 2D fits (Supplementary Fig. 3) also show a consistently larger intercept for TSBP than TNBP, resulting in higher $A$ values (Supplementary Fig. 6 and Supplementary Table 2). The $A$ values from the 3D fits are in the same range as those calculated for mixed primary/secondary ZDDPs ($10^{10}$−$10^{12}$ s$^{-1}$) using AFM[12]. However, much larger differences in $A$ were observed between primary ($6 \times 10^5$ s$^{-1}$) and secondary ($2 \times 10^{10}$ s$^{-1}$) ZDDPs in previous full-film EHL experiments[16]. This is probably because the primary (alkyl transfer) and secondary (β−H elimination) ZDDPs are believed to undergo different initial decomposition mechanisms to remove the alkyl groups[21], whereas the primary and secondary phosphate esters studied here both follow the same decomposition mechanism (C−O cleavage).

**Pre-exponential factor.** Since the pre-exponential factor seems to be driving the large differences in mechanochemical reactivity

between trialkylphosphates and ZDDPs with primary and secondary alkyl groups, it is important to understand its physical meaning. The pre-exponential factor in the Arrhenius equation is sometimes decomposed into $A = Z \rho$, where $Z$ is the frequency or collision factor, and $\rho$ is the steric factor, which is usually less than unity[14]. The rate of molecule−surface collisions ($Z$) has been quantified previously from NEMD simulations of solid−gas interfaces;[63] however, this is more challenging for solid−liquid interfaces. Given that their molecular structure only differs by the position of the C−O bond along the butyl chain, $Z$ is expected to be very similar for TNBP and TSBP. Previous experiments have shown almost identical adsorption behaviour for $n$-butyl and $s$-butyl ZDDPs from a hydrocarbon base oil onto iron oxide[64]. Therefore, the much higher reaction rates observed experimentally[16] and in the current NEMD simulations for additives containing secondary alkyl groups arise mostly from variations in $\rho$. This implies that the cleavage of the C−O bond in TNBP on the α-Fe(110) surface is more strongly dependent on the molecular conformation and a smaller fraction of molecule−surface collisions result in a reaction. Indeed, the through-film number density profile for the C atoms in TNBP (Supplementary Fig. 8a) have peaks that are split into two, one of which overlaps with the outer layer of Fe atoms that are pulled away from the surface during dissociative chemisorption[65]. This suggests that C−O cleavage reactions (Fig. 2a) only occur for TNBP when some of the C atoms become trapped in a specific orientation relative to the sliding surface. Conversely, the number density profile for the C atoms in TSBP (Supplementary Fig. 8b) shows that there is only one sharp peak at the periphery of each surface, suggesting that a larger proportion of molecule−surface collisions result in a C−O cleavage reaction (Fig. 2b), irrespective of the molecular conformation. This observation can be rationalised through the fact that the secondary carbocation or alkyl radical formed by C−O cleavage in TSBP will be much more stable and thus formed more readily than the primary equivalent formed from TNBP[49].

According to transition state theory, $A$ can also be expressed in terms of the activation entropy, $\Delta S^{\ddagger}$, of the reaction using the Eyring equation[66]. Given that the rate-determining step (C−O cleavage) is unimolecular, the rate constant can be calculated as,

$$k = \kappa \left(\frac{k_B T}{h}\right) \exp\left(-\frac{\Delta G^{\ddagger}}{RT}\right) = \kappa \left(\frac{k_B T}{h}\right) \exp\left(\frac{\Delta S^{\ddagger}}{R}\right) \exp\left(-\frac{\Delta H^{\ddagger}}{RT}\right)$$

(2)

where $h$ is Planck's constant, $R$ is the molar gas constant, $\kappa$ is the transmission factor (assumed to be unity), $\Delta G^{\ddagger}$ is the Gibbs free energy of activation, and $\Delta H^{\ddagger}$ is the activation enthalpy[54]. For SATA processes[13], $\Delta H^{\ddagger} = \Delta U^{\ddagger} - \sigma \Delta V^{\ddagger}$, where $\Delta U^{\ddagger}$ is the internal activation energy[67]. Using the 3D fits shown in Fig. 4c, we obtain values of $\Delta V^{\ddagger} = 16.4 \pm 2.7$ Å³, $\Delta H^{\ddagger} = 14.2 \pm 2.6$ kJ mol⁻¹ and $T \Delta S^{\ddagger} = -16.9 \pm 1.9$ kJ mol⁻¹ for TNBP at $T = 400$ K. From the 3D fits in Fig. 4d, $\Delta V^{\ddagger} = 16.7 \pm 4.0$ Å³, $\Delta H^{\ddagger} = 14.4 \pm 3.5$ kJ mol⁻¹ and $T \Delta S^{\ddagger} = -12.6 \pm 2.8$ kJ mol⁻¹ for TSBP at the same temperature. Negative entropy values are expected given that the phosphate esters are transformed from liquid molecules to surface-adsorbed species, which means that the transition state is more ordered than the reactants. The above values suggest that the smaller activation entropy penalty is mostly responsible for the higher reactivity of TSBP than TNBP. Previous DFT calculations have also highlighted the importance of entropic contributions to the mechanochemistry of phosphate esters[68]. Dissociative chemisorption for TNBP has a larger activation entropy penalty than TSBP, implying that the former has a more restricted transition state, in which translation and rotation are hindered[69]. This is due to the lower stability of the primary carbocation[49], which means that TNBP has a narrower

reaction path along the potential energy surface because there are fewer low-energy states in the vicinity of the transition state. Although the difference in activation entropy between TNBP and TSBP is relatively small (~4 kJ mol⁻¹), the exponential dependence of the rate on this quantity in the Eyring equation leads to much faster reactivity for TSBP under all of the conditions studied.

In summary, the higher mechanochemical reactivity of secondary TSBP compared to primary TNBP can be interpreted through different, although not contradictory, perspectives through either collision theory or transition state theory. Using the modified Arrhenius (Bell[15]) equation, the higher rate of TSBP than TNBP originates from a higher steric factor (and thus pre-exponential factor) for the former. Using the Eyring equation[66], the higher reactivity of TSBP can be attributed to a smaller activation entropy penalty, due to a less restricted transition state.

The general agreement between the mechanochemical parameters obtained from experiments and NEMD simulations with ReaxFF suggests that the latter are suitable for the virtual screening of antiwear additives with different alkyl substituents. This paves the way for the autonomous molecular design of new lubricant additives with optimised molecular structures for tailored tribological performance, as has recently been achieved for base oils[26]. To screen a large number of candidate molecular structures, significant acceleration of the NEMD simulations will be required, which could perhaps be achieved by utilising graphics processing units (GPUs)[70]. In addition to tribology, we anticipate that NEMD simulations with reactive force fields[44] will be useful to elucidate mechanochemical synthesis pathways by mimicking the conditions inside ball mills[1].

## Conclusions

We have used NEMD simulations with ReaxFF to compare the mechanochemical responses of primary (TNBP) and secondary (TSBP) trialkylphosphates heated and compressed between sliding iron surfaces. For both TNBP and TSBP, decomposition proceeds through dissociative chemisorption, during which one of the C−O bonds is broken. The rate of this process increases exponentially with temperature and shear stress, which implies that this is a SATA process. 2D fits to the Bell model suggest that the activation energy and pre-exponential factor both increase with pressure, which is indicative of a kinetic compensation effect. The activation volume increases with temperature, which can only partially be explained through a reduction in contact stiffness. 3D fits of the entire dataset to the Bell model over a wide range of temperature and stress conditions give reduced uncertainty in the model parameters compared to individual 2D fits. TSBP shows much faster dissociation rates than TNBP; however, both molecules have similar activation energy and activation volume. The much higher reactivity of TSBP is driven mostly by the pre-exponential factor, which is approximately four times larger than for TNBP. This is due to the higher stability of the secondary carbocation formed from the former following C−O cleavage. The additional stability enables a higher proportion of molecule−surface collisions to result in a reaction. This can also be interpreted as a smaller activation entropy penalty because of a less hindered transition state. Many of the observations from these NEMD simulations, as well as the parameters obtained from fits to the Bell model, are similar to those obtained from AFM and macroscale tribometer experiments using the ubiquitous antiwear additive ZDDP. The results provide further evidence that the initial dissociative chemisorption is the rate-determining step for tribofilm formation by antiwear additives. They also highlight the central role of the pre-exponential factor (or activation entropy) in distinguishing the mechanochemical reactivity of antiwear additives containing different alkyl groups. This study represents

an important step towards the virtual screening and autonomous design of antiwear additives to optimise their molecular structure for tailored mechanochemical and tribological responses.

## Methods

**System setup**. We compare the reactivity of two trialkylphosphates; TNBP, which contains primary linear alkyl groups and TSBP, which contains secondary alkyl groups. In addition to their primary application as antiwear additives for liquid lubricants[71], trialkylphosphates are also used as vapour phase lubricants[72]. The molecular structures of the two phosphate esters considered are shown in Fig. 1a. In all of the NEMD simulations, 48 trialkylphosphate molecules were randomly inserted between the sliding surfaces. No base oil molecules are considered in the simulations. Similar to full-film EHL tribometer experiments[16,18], no direct solid−solid contact occurs and the stress is applied through the confined molecules.

Surface analysis of steel surfaces following tribometer experiments using a lubricant with phosphorus-containing antiwear additive suggests that iron oxide is present at the surface[73]. However, during rubbing, nascent iron will be exposed, which will quickly react with phosphorus-containing antiwear additives[73]. To obtain sufficient reaction events in the accessible simulation time under experimentally-relevant conditions, we employed iron, rather than iron oxide surfaces. Previous ReaxFF MD simulations showed that, for TNBP and TSBP, the same initial thermal decomposition process (C−O cleavage) occurs on iron and iron oxide surfaces[27]. We selected the α-Fe(110) surface due to its higher thermodynamic stability than the other cleavage planes[74].

All of the systems were constructed using the Materials and Processes Simulations (MAPS) platform from Scienomics SARL. The two α-Fe(110) surfaces had dimensions of $x = 5.1$, $y = 4.8$ and $z = 1.1$ nm. Periodic boundary conditions are applied in the $x$ and $y$ directions. Before compression, the surfaces were initially separated by 4.0 nm of vacuum in the $z$-direction. A snapshot of a representative system is shown in Fig. 1b.

**Simulation procedure**. We used the large atomic/molecular massively parallel simulator (LAMMPS) software[75] for all of the NEMD simulations. We employed velocity Verlet integration[76] with a time step of 0.25 fs[46]. First, the systems were energy minimised using the conjugate gradient method. Equilibration simulations were then performed at an ambient temperature of 300 K and low pressure of 10 MPa for 0.1 ns. The temperature ($T = 300−500$ K), pressure ($\sigma_{zz} = 1−4$ GPa), and sliding velocity ($v_s = 10$ m s$^{-1}$) were then simultaneously increased to their target values. The selected temperature and pressure ranges align with those used in previous macroscale tribometer[16] and AFM experiments[12] of antiwear additive mechanochemistry. The temperature in the NEMD simulations was controlled with a Langevin thermostat[77] using a damping parameter of 25 fs. The thermostat was only applied to the middle layer of atoms in the slabs[78]. The pressure was increased by adding a constant normal force to the outer layer of atoms in the top slab, while the outer layer of atoms in the bottom slab was fixed in the $z$-direction[79]. The sliding velocity was imposed by adding equal and opposite velocities ($\pm5$ m s$^{-1}$) to the outer layer of atoms in the slabs in the $x$-direction (see Fig. 1b). These heating, compression, and shear simulations were performed for 1.0 ns, which was sufficient for the number of intact TNBP or TSBP molecules to decay to zero for most of the conditions studied. The temperature, pressure, and shear are commonly increased simultaneously in NEMD simulations of mechanochemistry[32,33], although some studies have separated these phases[35,36]. In most experimental studies of mechanochemistry of tribological systems[12,16–19], antiwear additives are dissolved in a lubricant, which is continuously entrained between the sliding surfaces. This means that new additive molecules become available to replenish those that have reacted with the sliding surfaces. In NEMD simulations, however, the number of antiwear additive molecules is finite. Therefore, instead of increasing them individually, we chose to increase the temperature, pressure, and sliding velocity concurrently to obtain an initial dissociation rate that captures all of these effects.

**Force field details**. The functional form of ReaxFF that is implemented in LAMMPS[75] was first outlined by Chenoweth et al.[80] and was described in more detail by Aktulga et al.[81]. The general form is given by[43]:

$$E_{ReaxFF} = E_{bond} + E_{over} + E_{angle} + E_{tors} + E_{vdW} + E_{Coulomb} + E_{specific} \quad (3)$$

where $E_{bond}$ is a continuous function of the interatomic distance and describes the energy associated with bond formation (including $\sigma$, $\pi$ and $\pi$-$\pi$ contributions). $E_{angle}$ and $E_{tors}$ are the energies associated with three-body angle and four-body torsional angle strain, respectively. $E_{over}$ is an energy penalty to prevent over-coordination of atoms and is based on atomic valence rules. $E_{Coulomb}$ and $E_{vdW}$ represent the electrostatic and dispersive interactions between all of the atoms in the system, irrespective of their connectivity and bond order. $E_{specific}$ represents system-specific terms required to capture particular properties of the system of interest, such as lone-pairs, conjugation and hydrogen bonding[43]. The point charges on the atoms vary dynamically during the MD simulation and are calculated using the charge equilibration (Qeq) method[81–83].

We employed the ReaxFF parameterisation developed for C/H/O/Fe/P-containing systems by Khajeh et al.[46]. They used the Fe/O/H parameters due to Aryanpour

et al.[84], the Fe/C parameters due to Zou et al.[85], the P/O/C/H parameters due to Verlackt et al.[86] and the C/H/O parameters due to Chenoweth et al.[80]. The ReaxFF parameters for Fe have been shown to accurately reproduce the experimental lattice parameters for α-Fe (within 1%)[85] and they have also been shown to perform favourably compared to other many-body force fields in describing its mechanical properties, including the elastic modulus[87]. The C/H/O/Fe/P ReaxFF parameterisation has recently been successfully applied to study the thermal decomposition of phosphate esters with different alkyl and aryl substituents on several ferrous surfaces[27,88]. The ReaxFF parameters have also been validated against DFT calculations for the adsorption energy and dissociation energy (including the energy barrier) for TNBP on α-Fe(110)[27]. Chemical bonding information was output every 1.0 ps, using a bond order cutoff of 0.3 to identify covalent bonds[80]. The choice of bond order cutoff only affects the post-processing analysis and does not influence the ReaxFF energy or force calculations during then NEMD simulations[27].

## Data availability

Data supporting the findings of this study are available within the article, the Supplementary Information, and the Supplementary Data file. The raw simulation data has been deposited in a public Zenodo repository available at: https://doi.org/10.5281/zenodo.5708426.

## Code availability

All of the MD simulations were performed using the open-source software LAMMPS, which is available under a GNU Public License Version 2 at: https://www.lammps.org/. The open-source software OVITO Basic was used for visualisation, which is available under the terms of the MIT License at: https://www.ovito.org/. The systems were constructed using the commercial MAPS software from Scienomics SARL (https://www.scienomics.com/).

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

## Acknowledgements

C.A.L. thanks the Engineering and Physical Sciences Research Council (EPSRC) and Afton Chemical for Ph.D. funding via the Theory and Simulation of Materials-Centre for Doctoral Training (TSM-CDT) EP/L015579/1. H.A.S., D.D., and J.P.E. thank the EPSRC for grant EP/P030211/1 and Established Career Fellowship EP/N025954/1. J.P.E. thanks the Royal Academy of Engineering for funding through a Research Fellowship. The authors acknowledge the use of Imperial College London Research Computing Service (https://doi.org/10.14469/hpc/2232) and the UK Materials and Molecular Modelling Hub, which is partially funded by EPSRC grants EP/P020194/1 and EP/T022213/1. We thank Jie Zhang for assistance with producing the 3D plots.

## Author contributions

J.P.E. designed the study, C.A.L. and J.P.E. performed the NEMD simulations, C.A.L. analysed the data and produced the figures, D.D., J.E.R., J.D.M. and J.P.E. supervised the research, D.D., H.A.S., and J.P.E. secured the funding, J.P.E. wrote the first draft of the manuscript, which was subsequently revised by all authors.

## Competing interests

The authors declare no competing interests.
