## [Peer Review File · Communications Chemistry]

Reviewers' comments:

Reviewer #1 (Remarks to the Author):

This is an interesting reactive-force-field MD study.

I have found only a few, non-central issues that can be dealt with in a revision:

1) lines 133-135: "For some systems, it has been argued that ReaxFF is incapable of accurately reproducing chemical reactions which are observed using first principles methods under tribological conditions."

There is no a-priori reason why a force field should not be able to capture every(!) result of an ab-initio or DFT calculation, regardless of which chemical system is studied. The Schrodinger equation has well-defined, unique, deterministic ground-state solutions, which are well-behaved functions of nuclear coordinates. This should be reproducible by a function that takes the same input as the ab-initio or DFT calculation (namely types of atoms and their coordinates, plus charge and spin state), is flexible enough and well parametrized.

Apparently, in the study cited (Ref.45), one or two given ReaxFF parameter sets did not lead to qualitatively good results. I would see this as a problem of parametrization, not of ReaxFF or of force fields in general.

Conversely, in that study, direct PBE-D dynamics were performed for reactive events. By chance, this may produce good results, but in general, bond breaking and bond formation events frequently involve electronically non-trivial situations, e.g., with significantly higher multireference character around the transition state than in the reactant and product minima. Hence, it cannot be expected that standard single-reference DFT with a standard functional will produce quantitatively reliable results in all cases. In such problematic cases, a force field can be fitted to multireference ab-initio data, and then may perform better than plain direct DFT.

Additionally, in direct ab-initio or DFT dynamics (without or with QM/MM partitioning), system sizes and/or trajectory length typically do not deliver sufficiently reliable statistics -- which is much less of a problem with force field dynamics.

In short, I would not recommend to yield so easily to one-sided arguments...

2) How did the authors arrive at 1.0 ns total simulated time, and why is this sufficient? Several figures extend somewhat beyond 1.0 ns.

3) lines 188-189: "Note that heterolysis and homolysis cannot be differentiated using the standard ReaxFF approach since electrons are not explicitly represented." Yes, coarse-graining away the electrons is the whole point of force fields (with eReaxFF being a weird exception). But to some extent, this difference should be visible in a force-field representation with non-constant atomic partial charges (as ReaxFF): In heterolytic bond cleavage, the fragments should be charged, while in homolytic cleavage they should be neutral. However, getting both options right, may be a real challenge for a force field (but not impossible in principle).

4) What is the "blue sphere" in Fig.2, c and d?

5) lines 345-348: "This implies that... and only a small fraction of molecule-surface collisions result in a reaction. Conversely, for TSBP, a much higher proportion..."

There is no need to speculate. In all-atom molecular dynamics, these data are directly available from

the trajectories. All that is required is a proper analysis. Then it will become directly obvious if these explanation attempts via bulk-level equations really are true at the atomistic level.

(Essentially, this is also true for the Eyring equation argument just below.)

I do understand that using such equations can be interpreted as a way to properly average the MD data, and as an interpretation aid, but after having arrived at an interpretation hypothesis, stepping back to the direct, atomistic level for verification would be the obvious thing to do, exploiting the actual strength of atomistic simulations, as compared to macroscopic measurements.)

Reviewer #2 (Remarks to the Author):

The authors carried out non-equilibrium molecular dynamics (NEMD) simulations of lubricants at iron/iron interfaces. They found that the TSBP molecules decompose faster than the TNBP molecules and that the reaction rates increase with an increasing contact stress. The authors fit a modified Arrhenius relationship to their reaction rates and reported that the difference in reaction rates is controlled “mostly by the pre-exponential factor” and that the difference in reaction energy barriers is small. Finally, the authors justified their findings in light of transition state theory (TST) as well as the collision theory.

I have a number of major concerns regarding this paper. First of all, the conclusion regarding the pre-exponential factor does not seem to be justified by the presented data. It is important to remember that the reaction rate depends linearly on the pre-exponential factor and exponentially on the energy barrier. The energy difference of 3kJ/mol (based on values shown in table 1) would result in a difference in reaction rates by a factor of 3.3 at 300K and a factor of 2 at 500K. The pre-exponentials differ by a factor of 5. This is assuming that ΔV^* doesn't change (as shown in Table 1). I don't see how the authors can argue that the difference in energy barriers is negligible compared to the role of the pre-exponential.

In addition, by combining the effect of the energy barrier and the pre-exponential factor, one would expect the difference in reaction rate to be ~ 10 at 500K and ~ 16 at 300K. However, by looking at Fig. 4, it seems that reaction rates at 500K differ by a factor of $\sim 110/40 = 2.8$ and at 300K maybe by a factor of 2? (it is difficult to read the data off the plot). Therefore, the values obtained from fitting to the Arrhenius relationship simply do not reproduce the data.

I suspect that the second issue mentioned above could result from the fact that the authors perform a 3D fit to data that involves different stresses and that there is a significant error bar on these fits. That brings me to the next issue that the authors should determine and report the error bars in their fits. The fitting procedure seems to involve subsequent fitting to data in Fig. 1 and then Fig. 2 of supplementary materials. The authors should report errors in those fits and propagate these errors. The factor of 5 in the pre-exponential term could be as well within the error bar of the measurement. Another issue with the 3D fits is that the activation volume ΔV^* is temperature independent and it is 29 \AA^3 for both molecules. However, from Fig. 4 it looks like ΔV^* actually increases linearly with an increasing temperature -- from 15 \AA^3 at 300K to 30 \AA^3 at 500K for TNBP and from 8 \AA^3 at 300K to 28 \AA^3 at 500K for TSBP. The use of the 3D fits does not seem to be justifiable here and the conclusions drawn from those fits seem to be on a shaky ground.

Assuming that the data fits are correct and conclusions are justified (which I'm not convinced of),

then the finding that the pre-exponential factor plays an important role is not particularly surprising. The authors cite an earlier experimental paper by Zhang et al. ACS Appl. Mater. Interfaces 12, 6662 (2020). In that earlier paper large differences in the pre-exponential factors were found in experiments on ZDDPs (slightly different but related lubricants) and a very similar analysis (in terms of TST and collision theory) was carried out. In the ACS Appl. Mater. Inter. Paper the difference in the pre-exponential factor was much more pronounced (4 orders of magnitude) and the analysis seemed much more robust than in the current paper. Therefore, it is not clear to me what are the new findings here that would justify publishing this manuscript in Comm. Chem.

Following up on the above discussion, the authors seem to compare results of their simulations to the previous experiments on ZDDPs molecule and say that the “kinetic parameters are consistent” between the simulations and experiments. I don’t see how the 4 orders of magnitude found experimentally are consistent with the factor of 5 found in simulations. The discussion of the comparison seems quite hand wavy.

Finally, the analysis in terms of TST and collision theory is qualitative and quite speculative. It is generally true that the Arrhenius relationship can be derived or related to these theories, which had been known. In the submitted manuscript, the authors do not actually calculate many of the parameters that enter the theories, but instead they speculate that for example “there is no obvious reason for a major difference in the rate of molecule-surface collision”. First of all, it’s not up to the reader to provide reasons for why that should not be true. Secondly, even small differences could explain the factor of 2-3 in reaction rates found in simulations so this type of qualitative discussion is not convincing.

Additional minor concerns:

- In Fig. 2, the colors of molecules seem to be mislabeled. Shouldn’t P the largest atom in the middle? Also, it is very difficult to understand what substituted for what based on the sequence of images in panel d. The panel is not very illustrative.
- Lines 345-346 read “the cleavage of the C–O bond in TNBP ... is strongly dependent on the molecular conformation”. However, in the conclusion part, lines 399-400 says “The higher mechanochemical reactivity of TSBP than TNBP is driven by the higher stability of the secondary carbocation ..., regardless of the molecular conformation”. It seems that these arguments are perhaps contradictory?

Reviewer #3 (Remarks to the Author):

- Equation 1 uses the symbol σ for stress, but the paragraph above it refers to σ_{zz} as the normal stress. Are these the same? Either way, the discrepancy should be clarified. I believe the difference between normal stress and shear stress is important in such processes, so addressing this point may require more than variable definition.
- The error bars on shear stress in Figures 1c and 1d are on the order of about 0.1 or 0.2 GPa. I assume there is similar error on the stress in the normal direction. It would be important to know this since it reflects how tightly controlled is the normal stress to the target value.
- How are bonds counted? Or, what criterion was used to determine if a bond was present or not between two atoms?
- I cannot see the difference between the various lines in Figure 2a and 2b. I think some are meant

to be darker than others, but the difference is very subtle, making it hard to interpret the plot.

- It is proposed that ΔV increases with temperature because of the temperature-dependent elasticity of the iron. However, the volume of material actually modelled and allowed to elastically deform is incredibly small. Is there any evidence that the very tiny volume of material has a change in elasticity? This can and should be checked.
- What is the physical meaning of A ?
- At what temperature are the values reported in Table 1 calculated?
- The fitted activation energies seem rather high, about ten times KT . Is this expected? It seems likely that bond dissociation energies could be found and compared to the fit value. Regardless, some discussion of this point is warranted.
- It is mentioned that the difference in A is more significant than the difference in E_a (between the two phosphate esters). However, the energy term is in an exponent. Can one be certain that the effect of A is in fact more significant?
- What is the physical meaning of the steric factor?
- Is the left-hand-side of Equation 3 supposed to be A ?
- I don't understand what is meant by "TNBP has a narrower reaction path". What does this mean? How was it determined? And, what is the implication?
- I am confused by the seemingly interchangeable use of the model names, Bell, Eyring, and Arrhenius.
- A better description and some proof should be provided for the "more hindered transition state" of TSBP.
- I do not understand this sentence: "we chose to increase the temperature, pressure, and sliding velocity concurrently to obtain a combined initial rate constant."
- This statement should be supported with simulation evidence: "The higher mechanochemical reactivity of TSBP than TNBP is driven by the higher stability of the secondary carbocation formed by the former following C–O cleavage".

We thank all three reviewers for their valuable and insightful comments. By addressing their comments in the revised manuscript, we feel that we have considerably improved it. Please find below a point-by-point response, with our replies highlighted in blue.

Reviewer #1

This is an interesting reactive-force-field MD study.

I have found only a few, non-central issues that can be dealt with in a revision:

1) lines 133-135: "For some systems, it has been argued that ReaxFF is incapable of accurately reproducing chemical reactions which are observed using first principles methods under tribological conditions."

There is no a-priori reason why a force field should not be able to capture every(!) result of an ab-initio or DFT calculation, regardless of which chemical system is studied. The Schroedinger equation has well-defined, unique, deterministic ground-state solutions, which are well-behaved functions of nuclear coordinates. This should be reproducible by a function that takes the same input as the ab-initio or DFT calculation (namely types of atoms and their coordinates, plus charge and spin state), is flexible enough and well parametrized.

Apparently, in the study cited (Ref.45), one or two given ReaxFF parameter sets did not lead to qualitatively good results. I would see this as a problem of parametrization, not of ReaxFf or of force fields in general.

Conversely, in that study, direct PBE-D dynamics were performed for reactive events. By chance, this may produce good results, but in general, bond breaking and bond formation events frequently involve electronically non-trivial situations, e.g., with significantly higher multireference character around the transition state than in the reactant and product minima. Hence, it cannot be expected that standard single-reference DFT with a standard functional will produce quantitatively reliable results in all cases. In such problematic cases, a force field can be fitted to multireference ab-initio data, and then may perform better than plain direct DFT.

Additionally, in direct ab-initio or DFT dynamics (without or with QM/MM partitioning), system sizes and/or trajectory length typically do not deliver sufficiently reliable statistics -- which is much less of a problem with force field dynamics.

In short, I would not recommend to yield so easily to one-sided arguments...

The reviewer makes an interesting point that we agree with. We originally mentioned and cited this study (Restuccia et al.) for balance, but we agree that the reported differences are more likely due to deficiencies in the force field parameterisation rather than the functional form of ReaxFF itself. We have clarified this by modifying this sentence to the following.

“For some systems, it has been argued that current ReaxFF parameterisations are incapable of accurately reproducing chemical reactions that are observed under tribological conditions using first principles methods”.

2) How did the authors arrive at 1.0 ns total simulated time, and why is this sufficient? Several figures extend somewhat beyond 1.0 ns.

We found that 1.0 ns was sufficient for the number of intact molecules to decay to zero for most of systems and conditions studied, as shown in Figure 3. In most of the figures, the 0.1 ns equilibration phase is also shown, so the entire simulation length is 1.1 ns. To clarify this, we have added the following sentence.

“These heating, compression, and shear simulations were performed for 1.0 ns, which was sufficient for the number of intact TNBP or TSBP molecules to decay to zero for most of the conditions studied.”

3) lines 188-189: "Note that heterolysis and homolysis cannot be differentiated using the standard ReaxFF approach since electrons are not explicitly represented."

Yes, coarse-graining away the electrons is the whole point of force fields (with eReaxFF being a weird exception). But to some extent, this difference should be visible in a force-field representation with non-constant atomic partial charges (as ReaxFF): In heterolytic bond cleavage, the fragments should be charged, while in homolytic cleavage they should be neutral. However, getting both options right, may be a real challenge for a force field (but not impossible in principle).

Since ReaxFF does only implicitly deal with electrons, the charge of different fragments is dependant upon the atomic coordinates and charge assignments from the Qeq algorithm. As such, a bond fission process will not yield easily-identifiable ions or radicals. Perhaps heterolytic and homolytic processes might go through different intermediate steps with different partial charges, but cataloguing them would be speculative, since ReaxFF does not explicitly differentiate them. A previous study,¹ which we have cited in the revised version (new Ref. 49), suggests that C-O cleavage is likely to be heterolytic.

4) What is the "blue sphere" in Fig.2, c and d?

The blue sphere is the P atom – this was incorrectly labelled in figure caption in the initial manuscript and this has been corrected. Many thanks for helping us spot this.

5) lines 345-348: "This implies that... and only a small fraction of molecule-surface collisions result in a reaction. Conversely, for TSBP, a much higher proportion..."

There is no need to speculate. In all-atom molecular dynamics, these data are directly available from the trajectories. All that is required is a proper analysis. Then it will become directly obvious if these explanation attempts via bulk-level equations really are true at the atomistic level.

(Essentially, this is also true for the Eyring equation argument just below.)

I do understand that using such equations can be interpreted as a way to properly average the MD data, and as an interpretation aid, but after having arrived at an interpretation hypothesis, stepping back to the direct, atomistic level for verification would be the obvious thing to do, exploiting the actual strength of atomistic simulations, as compared to macroscopic measurements.)

Whilst we agree with the reviewer that this is somewhat speculative; however, it is not straightforward to identify molecule-surface collisions for liquid-surface interface under compression and shear. Indeed, we are not aware of any previous MD simulations of similar systems that have successfully quantified this. However, looking at the number density profiles of the phosphate ester atoms between the surfaces, there are clear differences between TNBP and TSBP that support statement highlighted by the reviewer. We have added the following Supplementary Figure 4 and related text to the revised manuscript:

Supplementary Figure 4. Number density profiles for the C, O, and P atoms in the TNBP and TSBP molecules confined between the α -Fe(110) surfaces. Representative examples shown at 400 K and 2 GPa.

“Indeed, the number density profile for the C atoms in TNBP (Supplementary Figure 4a) have peaks that are split into two, one of which overlaps with the outer layer of Fe atoms that are pulled away from the surface during dissociative chemisorption.² This suggests that C–O cleavage reactions only occur for TNBP when some of the C atoms become trapped in a specific orientation relative to the surface. Conversely, the number density profile for the C atoms in TSBP (Supplementary Figure 4b) shows that there is only one sharp peak at the periphery of each surface, suggesting that a larger proportion of molecule–surface collisions result in a C–O cleavage reaction, irrespective of the molecular conformation. This observation can be rationalised through the fact that the secondary carbocation or alkyl radical formed by C–O cleavage in TSBP will be much more stable and thus formed more readily than the primary equivalent formed from TNBP.³”

Reviewer #2

The authors carried out non-equilibrium molecular dynamics (NEMD) simulations of lubricants at iron/iron interfaces. They found that the TSBP molecules decompose faster than the TNBP molecules and that the reaction rates increase with an increasing contact stress. The authors fit a modified Arrhenius relationship to their reaction rates and reported that the difference in reaction rates is controlled “mostly by the pre-exponential factor” and that the difference in reaction energy barriers is small. Finally, the authors justified their findings in light of transition state theory (TST) as well as the collision theory.

I have a number of major concerns regarding this paper. First of all, the conclusion regarding the pre-exponential factor does not seem to be justified by the presented data. It is important to remember that the reaction rate depends linearly on the pre-exponential factor and exponentially on the energy barrier. The energy difference of 3kJ/mol (based on values shown in table 1) would result in a difference in reaction rates by a factor of 3.3 at 300K and a factor of 2 at 500K. The pre-exponentials differ by a factor of 5. This is assuming that ΔV^* doesn't change (as shown in Table 1). I don't see how the authors can argue that the difference in energy barriers is negligible compared to the role of the pre-exponential.

We agree with the reviewer that the E_a can also play an important role and we have changed this discussion to better reflect the relative roles of the parameters in Equation 1 on the reaction rate. Note that a transposition error in Table 1, which was carried forward to the subsequent discussion, meant that the E_a and ΔV^* values were swapped in the initial manuscript. We apologise for this mistake that has been corrected in the revised manuscript. We also decided to perform additional NEMD simulations at high temperature and shear stress to complete the parameter space for the 2D and 3D fits to Equation 1. Therefore, all of the values in Table 1 have been updated. We have also made significant changes in the resulting discussion, as can be seen in the revised manuscript.

In addition, by combining the effect of the energy barrier and the pre-exponential factor, one would expect the difference in reaction rate to be ~ 10 at 500K and ~ 16 at 300K. However, by looking at Fig. 4, it seems that reaction rates at 500K differ by a factor of $\sim 110/40 = 2.8$ and at 300K maybe by a factor of 2? (it is difficult to read the data off the plot). Therefore, the values obtained from fitting to the Arrhenius relationship simply do not reproduce the data.

The transposition error in the Table 1 (values for E_a and ΔV^* were swapped) meant that the E_a values were incorrect. If we fix the transposition (without the new NEMD simulations), we get identical values for E_a and a 3 \AA^3 difference in ΔV^* (instead of identical ΔV^* and a difference of 3 kJ mol^{-1} in E_a). This difference in activation volumes would result in a factor of ~ 1.5 at 500 K and 1 GPa shear stress in favour of TNBP. Given the TSBP has a prefactor

that is ~5 times larger, TSBP would be expected to have $\sim 5/1.5 = 3.3$ times the reaction rate of TNBP under those conditions (a more detailed calculation shows it should be closer to 4.1). Indeed, in the 3D plots, you can see that the corresponding rates for TNBP and TSBP are $\sim 35 \text{ ns}^{-1}$ and $\sim 140 \text{ ns}^{-1}$, respectively.

For completeness, we have now added rates from additional NEMD simulations to complete the parameter space. We then repeated the fitting procedure to obtain the new kinetic constants for Equation 1 shown in Table 1. The new reaction constants are similar to the previous ones; nonetheless, the difference in absolute values of the reaction rates between TNBP and TSBP is now slightly more influenced by the difference in A .

Table 1. Calculated values of the activation energy, E_a , pre-exponential factor, A , and activation volume, ΔV^* , for TNBP (Figure 4c) and TSBP (Figure 4d) from the 3D fits. The parameter ranges represent the 95 % confidence intervals from the 3D fits.

	E_a (kJ mol ⁻¹)	A (s ⁻¹)	ΔV^* (Å ³)
TNBP	25 ± 6	$6.8 \pm 6.5 \times 10^{11}$	20 ± 6
TSBP	26 ± 5	$2.9 \pm 2.6 \times 10^{12}$	23 ± 5

Using the new data, for 500 K and 1 GPa we can calculate the difference in E_a would make TNBP ~1.4 times more reactive; conversely, the difference in ΔV^* would make TSBP ~1.4 times more reactive. Thus, both contributions approximately cancel out, leaving the prefactor A as the main cause for the difference in rates of a factor of 4.

I suspect that the second issue mentioned above could result from the fact that the authors perform a 3D fit to data that involves different stresses and that there is a significant error bar on these fits. That brings me to the next issue that the authors should determine and report the error bars in their fits. The fitting procedure seems to involve subsequent fitting to data in Fig. 1 and then Fig. 2 of supplementary materials. The authors should report errors in those fits and propagate these errors.

The authors agree with the reviewer that we should report the uncertainty in the parameters obtained from the 3D fits. These are now reported in Table 1 in the revised manuscript (see above).

The 3D fit was not obtained in a step-wise manner involving fits to the Figure 1 and then Supplementary Figure 2, as the reviewer suggests, but by fitting all of the results shown in the 3D plots to the Bell model (Equation 1) surface. While the shear stress values had some

inherent variability that could be calculated from the MD output (shown in the standard deviations of Figure 1), the uncertainty in the reaction rates were unknown. This is because the rates were obtained by fitting the number of intact phosphate ester molecules to a decaying exponential, as shown in Figure 3 (and Supplementary Figure 1), which yields a single rate for each condition. The variability in the rates could be estimated by performing several statistically-independent repeats of each of the conditions. However, this would be computationally prohibitive for the large number of systems and conditions studied here. Instead, we estimate the confidence bounds, c , for the parameters used in the fits to Equation 1 by:

$$c = a \pm t_{n,\alpha/2} \sqrt{S},$$

where a are the coefficients obtained in the fits, $t_{n,\alpha/2}$ is the Student's t score for n degrees of freedom and $1 - \alpha$ confidence level, and S are the diagonal elements from the estimated covariance matrix of the coefficients. These confidence bounds are shown in Table 1 in the revised manuscript (see above).

While the confidence bounds in the parameters from the 3D fits are quite wide, they are smaller than obtained for each of the individual 2D fits – see new Supplementary Table 1 below. The relatively large uncertainty in the 2D fits is because there are only 4 or 5 stress/temperature rate data points at each temperature/stress condition. On the other hand, the 3D surfaces are fit to 20 data points for both TNBP and TSBP. Sparse datasets are a limitation of both computational and experimental studies of mechanochemistry since significant effort is required to obtain a single rate result.

Supplementary Table 1. Calculated values of the activation energy, E_a , pre-exponential factor, A , and activation volume, ΔV^* , for TNBP and TSBP from the 2D fits. *using the mean ΔV^* from 300–500 K. The parameter ranges represent the 95 % confidence intervals from the 2D fits.

	$\Delta V^* [\text{\AA}^3]$			$A [\text{s}^{-1}]^*$				$E_a [\text{kJ mol}^{-1}]^*$			
	300 K	400 K	500 K	1 GPa	2 GPa	3 GPa	4 GPa	1 GPa	2 GPa	3 GPa	4 GPa
TNBP	15 ± 8	20 ± 4	29 ± 5	$2 \pm 2 \times 10^{10}$	$7 \pm 4 \times 10^{10}$	$1 \pm 1 \times 10^{11}$	$1 \pm 1 \times 10^{11}$	11 ± 4	16 ± 4	18 ± 5	21 ± 9
TSBP	8 ± 3	19 ± 9	28 ± 9	$4 \pm 4 \times 10^{10}$	$1 \pm 1 \times 10^{11}$	$6 \pm 6 \times 10^{11}$	$1 \pm 1 \times 10^{12}$	11 ± 7	15 ± 7	20 ± 9	22 ± 9

Previous tribology experiments of mechanochemistry have rarely reported the uncertainty in the parameters obtained from 2D fits to the Bell model. When the confidence intervals have

been reported, as in the seminal paper by Gosvami et al. *Science*. 348, 102-106 (2015), they are comparable to those for ΔV^* and E_a ($\pm 30\%$) in the current study. Moreover, in the Supporting Information, Gosvami et al. state that they can fit their data with the prefactor A ranging from 10^{10} to 10^{12} s^{-1} , whereas this parameter was also included in our fitting procedure. Thus, even in our 2D fits, we obtain a much smaller uncertainty for A (at the cost of increased uncertainty in the other two parameters).

As further proof that the rates observed in the NEMD simulations can be adequately reproduced by the rates obtained with the Bell model fitted to the 3D surface, they are plotted against each other in Supplementary Figure 3 with satisfactory R^2 values:

Supplementary Figure 3. Comparison of the reaction rates calculated from the NEMD simulations and predicted using Equation 1. $R^2 = 0.85$ for TNBP and $R^2 = 0.89$ for TSBP.

The factor of 5 in the pre-exponential term could be as well within the error bar of the measurement. Another issue with the 3D fits is that the activation volume ΔV^* is temperature independent and it is 29 \AA^3 for both molecules. However, from Fig. 4 it looks like ΔV^* actually increases linearly with an increasing temperature -- from 15 \AA^3 at 300K to 30 \AA^3 at 500K for TNBP and from 8 \AA^3 at 300K to 28 \AA^3 at 500K for TSBP. The use of the 3D fits does not seem to be justifiable here and the conclusions drawn from those fits seem to be on a shaky ground.

As explained above, the 3D fitting is actually a more robust approach to fit the entire dataset and extract E_a , A , and ΔV^* values compared to the 2D fitting approach we (Zhang et al. *ACS Appl. Mater. Interfaces* 12, 6662 (2020)) and others (e.g. Gosvami et al. *Science*. 348, 102-106 (2015)) have used previously for experimental mechanochemistry data.

Our results yield uncertainties of up to 10 \AA^3 in the activation volume (Supplementary Table 1) when they are calculated from the slope of the logarithm of the reaction rate as a function of shear stress (insets of Figure 4). It can be concluded that activation volumes obtained in this manner, as often done experimentally, can be subject to substantial variability. The 3D fits, on the other hand, feed the Bell model more data and are, in our opinion, both more robust and more representative of the wide range of conditions that lubricant additives are subjected to inside rubbing contacts.

Assuming that the data fits are correct and conclusions are justified (which I'm not convinced of), then the finding that the pre-exponential factor plays an important role is not particularly surprising. The authors cite an earlier experimental paper by Zhang et al. *ACS Appl. Mater. Interfaces* 12, 6662 (2020). In that earlier paper large differences in the pre-exponential factors were found in experiments on ZDDPs (slightly different but related lubricants) and a very similar analysis (in terms of TST and collision theory) was carried out. In the *ACS Appl. Mater. Inter.* Paper the difference in the pre-exponential factor was much more pronounced (4 orders of magnitude) and the analysis seemed much more robust than in the current paper. Therefore, it is not clear to me what are the new findings here that would justify publishing this manuscript in *Comm. Chem.*

As described in the responses above above, the analysis methods chosen are robust. We have significantly changed the discussion regarding the relative roles of E_a , A , and ΔV^* based on the updated values in Table 1.

The *ACS Appl. Mater. Interfaces* paper concerned experiments on ZDDP. There are a number of important differences between that paper (and anything else we are aware of in the literature) and the current manuscript. Firstly, while related, phosphate esters and ZDDP are different compounds and therefore subject to different mechanochemical decomposition mechanisms. Elucidating the mechanochemistry of phosphate ester decomposition will serve both to understand these compounds and highlight the similarities and differences with ZDDP. As explained in the revised manuscript, unlike ZDDPs, phosphate esters with primary and secondary substituents, do not decompose through different mechanisms. Therefore, the differences in reactivity between TNBP and TSBP are more subtle (although still significant). Our NEMD simulations allow us to directly observe the bond fission and formation processes and they suggest that C-O cleavage is favoured over P-O cleavage or β -H elimination for both TNBP and TSBP on sliding iron surfaces.

To the best of our knowledge, our study represents the first instance that the Bell model has been fitted to a 3D surface corresponding to a wider range of conditions, with the corresponding improvement in the uncertainty in the reaction constants, as evidenced

above. Following similar modelling approaches in the future can provide a new and robust technique to investigate the stress-assisted reactivity of a wide range of compounds. The conditions used in this type of NEMD simulation can be similar to those inside tribometers (as in the current study) or ball mills used for mechanochemical synthesis. This opens a wide range of possibilities of elucidating mechanochemical reaction pathways and the associated kinetic parameters using NEMD simulations as a virtual screening tool.

Following up on the above discussion, the authors seem to compare results of their simulations to the previous experiments on ZDDPs molecule and say that the “kinetic parameters are consistent” between the simulations and experiments. I don't see how the 4 orders of magnitude found experimentally are consistent with the factor of 5 found in simulations. The discussion of the comparison seems quite hand wavy.

We agree with the reviewer and this discussion has changed considerably due to the corrected E_a and ΔV^* values. Nonetheless, both our simulations and the experiments find that the prefactors are the main difference between primary and secondary substituents in these additives, which is an important discovery. As explained above, the reaction constants obtained in linear fits such as in the ZDDP experiments are subject to large uncertainties; the prefactor A is particularly subject to large uncertainties, e.g. the Gosvami *et al.* paper that reported good fits with prefactors spanning several orders of magnitude. We have clarified the reasons why the differences in A are smaller for phosphates than ZDDPs.

Finally, the analysis in terms of TST and collision theory is qualitative and quite speculative. It is generally true that the Arrhenius relationship can be derived or related to these theories, which had been known. In the submitted manuscript, the authors do not actually calculate many of the parameters that enter the theories, but instead they speculate that for example “there is no obvious reason for a major difference in the rate of molecule-surface collision”. First of all, it's not up to the reader to provide reasons for why that should not be true. Secondly, even small differences could explain the factor of 2-3 in reaction rates found in simulations so this type of qualitative discussion is not convincing.

Whilst we agree with the reviewer that this is somewhat speculative, it is not straightforward to identify molecule-surface collisions for liquid-surface interface under compression and shear. Indeed, we are not aware of any previous MD simulations of similar systems that have successfully quantified this. However, looking at the number density profiles of the phosphate ester atoms between the surfaces, there are clear differences between TNBP and TSBP that support statement highlighted by the reviewer. We have added the following Supplementary Figure 4 and related text to the revised manuscript.

Supplementary Figure 4. Number density profiles for the C, O, and P atoms in the TNBP and TSBP molecules confined between the α -Fe(110) surfaces. Representative examples shown at 400 K and 2 GPa.

“Indeed, the number density profile for the C atoms in TNBP (Supplementary Figure 4a) have peaks that are split into two, one of which overlaps with the outer layer of Fe atoms that are pulled away from the surface during dissociative chemisorption.² This suggests that C–O cleavage reactions only occur for TNBP when some of the C atoms become trapped in a specific orientation relative to the surface. Conversely, the number density profile for the C atoms in TSBP (Supplementary Figure 4b) shows that there is only one sharp peak at the periphery of each surface, suggesting that a larger proportion of molecule–surface collisions result in a C–O cleavage reaction, irrespective of the molecular conformation. This observation can be rationalised through the fact that the secondary carbocation or alkyl radical formed by C–O cleavage in TSBP will be much more stable and thus formed more readily than the primary equivalent formed from TNBP.³”

Additional minor concerns:

- In Fig. 2, the colors of molecules seem to be mislabeled. Shouldn't P the largest atom in the middle? Also, it is very difficult to understand what substituted for what based on the sequence of images in panel d. The panel is not very illustrative.

The reviewer is correct; thanks for spotting this – the mislabelling of the P and O atoms has also been corrected on both Figure 1 and Figure 2.

- Lines 345-346 read “the cleavage of the C–O bond in TNBP ... is strongly dependent on the molecular conformation”. However, in the conclusion part, lines 399-400 says “The higher mechanochemical reactivity of TSBP than TNBP is driven by the higher stability of the secondary carbocation ..., regardless of the molecular conformation”. It seems that these arguments are perhaps contradictory?

These sentences may have been unclear in the initial manuscript and so they have been revised. TNBP requires a specific molecule-surface conformation for C–O dissociation to occur, whereas for TSBP the reaction proceeds irrespective of the molecular conformation.

Reviewer #3

- Equation 1 uses the symbol σ for stress, but the paragraph above it refers to σ_{zz} as the normal stress. Are these the same? Either way, the discrepancy should be clarified. I believe the difference between normal stress and shear stress is important in such processes, so addressing this point may require more than variable definition.

σ is used as a general stress term, σ_{zz} is the normal stress and σ_{xy} is the shear stress.

- The error bars on shear stress in Figures 1c and 1d are on the order of about 0.1 or 0.2 GPa. I assume there is similar error on the stress in the normal direction. It would be important to know this since it reflects how tightly controlled is the normal stress to the target value.

The error bars for the normal stress in Figure 1c and 1d are smaller than the symbol size. This information has been added to the figure caption.

- How are bonds counted? Or, what criterion was used to determine if a bond was present or not between two atoms?

This description was included in the Methodology,

“Chemical bonding information was output every 1.0 ps, using a bond order cutoff of 0.3 to identify covalent bonds.⁴ The choice of bond order cutoff only affects the post-processing analysis and does not influence the ReaxFF energy or force calculations.⁵”

This is standard practice in ReaxFF MD simulations. We have previously performed sensitivity analysis on the bond order cutoff (± 0.1) and found that although the overall number of bonds increases/decreases, the changes are small and the trends between the different systems and conditions remains the same.

- I cannot see the difference between the various lines in Figure 2a and 2b. I think some are meant to be darker than others, but the difference is very subtle, making it hard to interpret the plot.

We appreciate that the dark and light lines looked similar, so we have changed to solid and dashed lines instead in the revised manuscript.

- It is proposed that ΔV increases with temperature because of the temperature-dependent elasticity of the iron. However, the volume of material actually modelled and allowed to elastically deform is incredibly small. Is there any evidence that the very tiny volume of material has a change in elasticity? This can and should be checked.

This was proposed based on the recent work of Li and Szlufarska,⁶ who found using DFT calculations of silica surfaces that the activation volume was inversely proportional to the interfacial stiffness. Previous experiments have shown that elastic modulus of iron decreases by approximately 10 % when the temperature is increased from 300 to 500 K.⁷

The ReaxFF parameters we used accurately reproduce the elastic modulus for bulk iron at 300 K.⁸ We also confirmed that our thin slabs they reproduce the bulk elastic modulus and that this decreases by ~10 % as T increases from 300 to 500 K. However, since the two properties are inversely proportional according to Li and Szlufarska,⁶ this small change in elastic modulus cannot explain the large change in ΔV^* . We have acknowledged this in the revised manuscript.

- What is the physical meaning of A?

The pre-exponential factor in the Arrhenius equation is sometimes decomposed into $A = Z \rho$, where Z is the frequency or collision factor, and ρ is the steric factor, which is usually less than unity. In this context, A is often interpreted as the frequency of vibration of a bond that attempts to dissociate, and the exponential term in the Arrhenius equation carries the probability for that dissociation to happen. As opposed to the empirically derived Arrhenius model, the similar Eyring model derives what would be A as a function of the transmission factor (assumed unity) and the activation entropy.

- At what temperature are the values reported in Table 1 calculated?

The values in Table 1 are global fit parameters for all of the temperature and pressure conditions studied. Separate values for 300-500 K are given in Supplementary Table 1.

- The fitted activation energies seem rather high, about ten times kT. Is this expected? It seems likely that bond dissociation energies could be found and compared to the fit value. Regardless, some discussion of this point is warranted.

Yes, these are broadly the sorts of values that are expected for mechanochemical dissociation of antiwear additives – see the comparisons to experimental values for ZDDP that are somewhat higher than those calculated here for phosphate esters. In any case, ten times kT yields an exponential factor of $\exp(-10) \sim 5 \times 10^{-5}$; multiplied by prefactors on the order of 10^{12} s^{-1} , they result in reasonable reaction rates.

- It is mentioned that the difference in A is more significant than the difference in E_a (between the two phosphate esters). However, the energy term is in an exponent. Can one be certain that the effect is A is in fact more significant?

We apologise that the E_a and ΔV^* values were mixed up in Table 1 in the original submission. After fixing the transposition, and with slight changes in the parameters as a result of more simulations run (see the updated Table 1 in the manuscript), for 500 K and 1 GPa we can calculate the difference in energy barriers would make TNBP ~1.4 times more reactive; conversely, the difference in activation volumes would make TSBP ~1.4 times more reactive. Thus, both contributions approximately cancel out, leaving the prefactors as the main cause for the difference in rates.

- What is the physical meaning of the steric factor?

$A = Z \rho$, where Z is the frequency or collision factor, and ρ is the steric factor: ρ can be interpreted as the ratio between the reactive collision cross section and the total collision cross section. In other words, a smaller steric factor is expected for molecule-surface combinations that require a specific orientation for a reaction to occur.

- Is the left-hand-side of Equation 3 supposed to be A ?

No, the equation is correct. There is no A in the Eyring equation (instead uses ΔS^\ddagger).

- I don't understand what is meant by "TNBP has a narrower reaction path". What does this mean? How was it determined? And, what is the implication?

Given that the difference in reactivities between TNBP and TSBP, and the fact that they seem to follow the same reaction mechanism, the Eyring model indicates the difference lays in the activation entropy difference; that is, in the probabilities of TNBP and TSBP molecules of finding themselves around the transition state. A narrower reaction path in the potential energy surface means a higher activation entropy and vice versa. This can also be understood as a narrower potential well.

- I am confused by the seemingly interchangeable use of the model names, Bell, Eyring, and Arrhenius.

These equations are related but not interchangeable; in the revised manuscript, we believe that we are clear about which we use when and why.

- A better description and some proof should be provided for the "more hindered transition state" of TSBP.

We have rephrased this to:

"Dissociative chemisorption for TNBP has a larger activation entropy penalty than TSBP, implying that the former has a more restricted transition state, in which translation and rotation are hindered.⁹ This is due to the lower stability of the primary carbocation,³ which

means that TNBP has a narrower reaction path along the potential energy surface because there are fewer low-energy states in the vicinity of the transition state.”

• I do not understand this sentence: “we chose to increase the temperature, pressure, and sliding velocity concurrently to obtain a combined initial rate constant.”

We have rephrased this to:

“Therefore, instead of increasing them individually, we chose to increase the temperature, pressure, and sliding velocity concurrently to obtain an initial dissociation rate that captures all of these effects.”

• This statement should be supported with simulation evidence: “The higher mechanochemical reactivity of TSBP than TNBP is driven by the higher stability of the secondary carbocation formed by the former following C–O cleavage”.

It is well known that secondary phosphate ester carbocations are more stable than primary ones.³ All of the simulation evidence presented shows that TSBP dissociates more readily than TNBP – and this is the key structural difference between the two molecules.

References

1. Higgins, C. E. & Baldwin, W. H. The Thermal Decomposition of Tributyl Phosphate. *J. Org. Chem.* **26**, 846–850 (1961).
2. Gattinoni, C., Ewen, J. P. & Dini, D. Adsorption of Surfactants on α -Fe₂O₃(0001): A Density Functional Theory Study. *J. Phys. Chem. C* **122**, 20817–20826 (2018).
3. Loncke, P. G. & Berti, P. J. Implications of protonation and substituent effects for C–O and O–P bond cleavage in phosphate monoesters. *J. Am. Chem. Soc.* **128**, 6132–6140 (2006).
4. Chenoweth, K., van Duin, A. C. T. & Goddard III, W. A. ReaxFF reactive force field for molecular dynamics simulations of hydrocarbon oxidation. *J. Phys. Chem. A* **112**, 1040–1053 (2008).
5. Ewen, J. P. *et al.* Substituent Effects on the Thermal Decomposition of Phosphate Esters on Ferrous Surfaces. *J. Phys. Chem. C* **124**, 9852–9865 (2020).
6. Li, Z. & Szlufarska, I. Physical Origin of the Mechanochemical Coupling at Interfaces. *Phys. Rev. Lett.* **126**, 076001 (2021).
7. Adams, J. J., Agosta, D. S., Leisure, R. G. & Ledbetter, H. Elastic constants of monocrystal iron from 3 to 500 K. *J. Appl. Phys.* **100**, 113530 (2006).

8. Morrissey, L. S., Handrigan, S. M., Subedi, S. & Nakhla, S. Atomistic uniaxial tension tests: investigating various many-body potentials for their ability to produce accurate stress strain curves using molecular dynamics simulations. *Mol. Simul.* **45**, 501–508 (2019).
9. Bucko, T. & Hafner, J. Entropy effects in hydrocarbon conversion reactions: Free-energy integrations and transition-path sampling. *J. Phys. Condens. Matter* **22**, 384201 (2010).

Reviewers' comments:

Reviewer #1 (Remarks to the Author):

I am satisfied with the answers and MS changes by the authors.

Reviewer #2 (Remarks to the Author):

The authors responded to some of our questions, but unfortunately there are a few important concerns that remain.

We originally had a concern that the difference in prefactors may not be that large as compared to the difference in the energy barriers. The authors replied that the original values of fitted data were wrong, and with the new data, the differences in energy barriers and activation volumes between the two molecules cancel out. Consequently, according to the authors, the difference in prefactors is the main reason for a variation in reaction rates. However, to demonstrate the cancellation of terms, the authors performed calculations for one specific condition, i.e., simulations at 500K and 1 GPa. This argument may not work if different conditions (from the authors' simulations) are considered. For instance, since the activation volume ΔV affects the energy barrier through a term $\sigma\Delta V$, if the pressure σ is reduced by half, then the contribution from the difference in the activation volume will be half of what the authors calculated. As a result, the effects of the intrinsic barrier and of the $\sigma\Delta V$ term will not cancel out. Considering that the difference in prefactors is only about a factor of ~ 4 , the statement that the prefactor difference dominates the difference in reaction rate does not have a strong support.

In the original review we were also concerned that the authors did not report error bars on their plots. In the revised version, the authors provide the error bars for both 2D and 3D fittings. According to the authors, the variations in the fitted values from 2D fitting are much larger than that of 3D fitting, implying that the 3D fitting is a more robust way to obtain the prefactor, activation volume, and energy barrier from the simulations. One problem with this analysis is that the variation in 2D fits at different conditions has a clear trend, which means that it is likely not a random error. This could be an indication that there is a hidden physical trend that cannot be captured by the single Bell model (Eq. 1). From supplementary table 1, it can be observed that the activation volume increases monotonically with temperature, and the prefactor and energy barrier also show increasing trend as a function of pressure. The authors actually acknowledge that the temperature-dependent activation volume could be physical in the main text. 3D fitting becomes questionable if these trends are indeed physical.

Even if there are no such physical trends and the variation in 2D fitting is random in nature, we have another serious concern about the 3D fit. Specifically, the 3D fitting is biased towards the largest data in the set. For instance, the prefactor fitted by 2D fitting varies by as much as ~ 2 orders of magnitude for TSBP with the largest prefactor of $\sim 10^{12}$ at 4 GPa. The 3D fitting gives a similar prefactor of $\sim 10^{12}$. These results are a manifestation of the fact that the 3D fitting is dominated by the data at high pressure (4 GPa). If we only look at the 2D fitting error at 4 GPa, which is 1×10^{12} , it is comparable to (even smaller than) than 3D fitting's error, which is 2.6×10^{12} . That means that the fitting error in 2D is not larger than in 3D as claimed by the authors. More importantly, the authors used one single value to represent pre-factors that vary over 2 orders of magnitude and then they

argue that the four-fold difference in the pre-factors between the molecules is meaningful. This does not seem to make much sense. If the two orders of magnitude in pre-factors result from uncertainty in the measurement, then the factor of 4 is meaningless.

Originally, we also had the question on the novelty of the current manuscript, compared to the previous paper of Zhang et al. *ACS Appl. Mater. Interfaces* 12, 6662 (2020). The authors pointed out two differences between the current manuscript and the published paper. One is that these two papers are studying different compounds with different decomposition mechanism. We leave it to the editors to decide whether this is a significant enough result to publish in this journal. The second claim to novelty is that this the first attempt of 3D fitting using Bell's model. As explained above, we think it is still not very convincing that 3D fitting is superior to 2D fitting or that such fit is even reasonable for the data obtained from current simulations.

Finally, we were also not convinced by the discussion of the transition state theory and collision theory presented in the original manuscript, as the authors did not provide supports of the theory from simulations. In the revised version, the authors provided additional simulation results, i.e., Supplementary Figure 4., which show the number density profiles of the phosphate ester atoms between the surfaces. However, the interpretation of these plots is unclear. According to the authors, the peak of the number density profile for C in TNBP near the surface is split, whereas the number density profile of C in TSBP has a single peak near the surface. The authors write that the split peak of TNBP is an indication that C-O cleavage occurs only for some fraction of C atoms. The single peak of TSBP according to the authors implies that a large fraction of molecular collisions has led to C-O cleavage. However, as shown in Supplementary Fig. 4 (b), the single C peak of TSBP near the surface overlaps with O peak, while the split peak of TNBP in (b) does not overlap of O peak. This seems to support the opposite conclusion, i.e., there are more C-O cleavage for TNBP. Either way, the authors should justify better their conclusions from the density profiles presented in the revised manuscript.

Reviewer #3 (Remarks to the Author):

Comments mostly addressed

Reviewer #2 (Remarks to the Author):

The authors responded to some of our questions, but unfortunately there are a few important concerns that remain.

We have carefully considered the remaining points raised by the reviewer, which they felt were not fully addressed through our initial revision. We have made several additional changes to the revised manuscript, which we are confident should address their concerns. The details of these changes are described below.

We originally had a concern that the difference in prefactors may not be that large as compared to the difference in the energy barriers. The authors replied that the original values of fitted data were wrong, and with the new data, the differences in energy barriers and activation volumes between the two molecules cancel out. Consequently, according to the authors, the difference in prefactors is the main reason for a variation in reaction rates. However, to demonstrate the cancellation of terms, the authors performed calculations for one specific condition, i.e., simulations at 500K and 1 GPa. This argument may not work if different conditions (from the authors' simulations) are considered. For instance, since the activation volume ΔV^\ddagger affects the energy barrier through a term $s\Delta V^\ddagger$, if the pressure s is reduced by half, then the contribution from the difference in the activation volume will be half of what the authors calculated. As a result, the effects of the intrinsic barrier and of the $s\Delta V^\ddagger$ term will not cancel out. Considering that the difference in prefactors is only about a factor of ~ 4 , the statement that the prefactor difference dominates the difference in reaction rate does not have a strong support.

While the activation volume and activation energy contributions discussed in the previous reply tend to cancel out, we agree with the reviewer that, with the previous parameters, this may depend on the conditions. However, based on the reviewer's suggestions, we have now performed a new 3D fitting procedure (based on the logarithm of the rates) that is not biased by the highest rates (see details below). This method conclusively shows that the pre-exponential factor (A) dominates the difference in mechanochemical decomposition rates between TNBP and TSBP.

In the original review we were also concerned that the authors did not report error bars on their plots. In the revised version, the authors provide the error bars for both 2D and 3D fittings. According to the authors, the variations in the fitted values from 2D fitting are much larger than that of 3D fitting, implying that the 3D fitting is a more robust way to obtain the prefactor, activation volume, and energy barrier from the simulations. One problem with this analysis is that the variation in 2D fits at different conditions has a clear trend, which means that it is likely not a random error. This could be an indication that there is a hidden physical

trend that cannot be captured by the single Bell model (Eq. 1). From supplementary table 1, it can be observed that the activation volume increases monotonically with temperature, and the prefactor and energy barrier also show increasing trend as a function of pressure. The authors actually acknowledge that the temperature-dependent activation volume could be physical in the main text. 3D fitting becomes questionable if these trends are indeed physical.

To investigate this point, we have added new Supplementary Figures 4-6 (below) to show how the various parameters (and 95% confidence intervals) change from the 2D fits and a comparison to the value obtained from the 3D fits. These also include values of dV calculated at 350K and 450K, which were not included in original SI Table 1. Including these values shows that the activation volume does not increase monotonically with temperature, as shown below. However, there is a general increase in the parameter, which may give some additional physicochemical insights. We have discussed this in detail in the revised manuscript (Pages 12 and 13).

Similarly, as shown below, E_a and $\ln(A)$, also show a general increase with temperature. This is discussed in the revised manuscript (Page 13).

As shown below, there is a clear linear relationship between the 2D fit values of $\ln(A)$ and E_a (see below), which is indicative of the kinetic compensation effect (<https://doi.org/10.1039/C1CP22666E>). With this in mind, and given the size of the error bars, it is unlikely that any physicochemical insights from the concurrent increase in $\ln(A)$ and E_a with increasing pressure. It is more likely that the increase in both parameters is mathematical in origin (<https://doi.org/10.1039/C1CP22666E>). This has been clarified in the revised manuscript.

As for any empirical relationship, the Arrhenius (and Bell) equation has several well-known limitations. E_a is a simplified average of all the processes taking place, and not just a single energy barrier through a predefined pathway as is sometimes discussed. Equivalently, the probability of reaction is a complex function of internal as well as translational energy, chemical environment, etc., and the pre-exponential factor encapsulates what is often understood as an attempt frequency for a reaction to take place, which in turn may be dependant on T and the reactant density (i.e. pressure). Some studies have focused explicitly on these aspects. See, for example, Menzinger and Wolfgang (<https://doi.org/10.1002/anie.196904381>), where different relationships between measured activation energies and temperature are discussed. For more concrete examples, see the case study of deep level emission in Cu(In,Ga)Se₂ solar cells by Li et al., where they found large variability of E_a with T (Fig 1 of <https://doi.org/10.1063/1.3361130>); or the observed variability of E_a with P and dV with T in the inactivation of avocado polyphenoloxidase by Weemaes et al. ([https://doi.org/10.1002/\(sici\)1097-0290\(19981105\)60:3<292::aid-bit4>3.0.co;2-c](https://doi.org/10.1002/(sici)1097-0290(19981105)60:3<292::aid-bit4>3.0.co;2-c)).

These studies (and others already cited in the manuscript) often identify systematic variability of the reaction “constants”. However, these parameters were obtained with 2D fits to the Arrhenius equation, which pre-assumes that they are constants. Indeed, to obtain E_a from the slope of $\ln(k)$ vs. $1/T$, one must assume that A is independent of T (and similarly with dV and P in the stress-augmented Arrhenius equation). However, these variables might be strictly be written $A(T,s)$, $E_a(T,s)$, $dV(T,s)$.

In the current study, it was not our aim to investigate the underlying physical factors behind the environmental variability of the reaction “constants”. Instead, we found the modified Arrhenius (Bell) model describes our parameter space (which is more comprehensive than other reports in the literature) quite accurately. This has been demonstrated in the revised manuscript and is further expanded upon below. In this context, the 3D fits suits our main purpose since they encompass the whole range of conditions simultaneously, providing us

with values that hold over this wide range and allow us to explain the difference in TNBP and TSBP reaction rates (i.e. their pre-exponential factors). On the other hand, the individual 2D fits generally have larger uncertainties and cannot be expected to represent reactions occurring outside of the specific set of conditions under which they were calculated.

Even if there are no such physical trends and the variation in 2D fitting is random in nature, we have another serious concern about the 3D fit. Specifically, the 3D fitting is biased towards the largest data in the set. For instance, the prefactor fitted by 2D fitting varies by as much as ~2 orders of magnitude for TSBP with the largest prefactor of $\sim 10^{12}$ at 4 GPa. The 3D fitting gives a similar prefactor of $\sim 10^{12}$. These results are a manifestation of the fact that the 3D fitting is dominated by the data at high pressure (4 GPa). If we only look at the 2D fitting error at 4 GPa, which is 1×10^{12} , it is comparable to (even smaller than) than 3D fitting's error, which is 2.6×10^{12} . That means that the fitting error in 2D is not larger than in 3D as claimed by the authors. More importantly, the authors used one single value to represent pre-factors that vary over 2 orders of magnitude and then they argue that the four-fold difference in the pre-factors between the molecules is meaningful. This does not seem to make much sense. If the two orders of magnitude in pre-factors result from uncertainty in the measurement, then the factor of 4 is meaningless.

We agree with the reviewer and thank them for their useful suggestion. It is correct that our 3D fitting in the last two versions of the manuscript was heavily biased towards the largest rates. In this version of the revised manuscript, we have re-fit the data on a logarithmic basis. This essentially replicates the approach used for the 2D fits. Hence, the residuals are no longer dominated by a few extreme values, which results in a large reduction in the uncertainty in the 3D fits. The new values, as included in the text, are:

	E_a (kJ mol ⁻¹)	ln(A) (s ⁻¹)	ΔV^* (Å ³)
TNBP	17.4 ± 2.6	25.6 ± 0.6	16.4 ± 2.8
TSBP	17.5 ± 3.6	26.9 ± 0.9	16.7 ± 4.0

The new values clearly indicate that the prefactors dominate the differences in the reaction rates. Also, the uncertainties in the 3D fits are now generally much lower than the 2D fits.

We also performed equal variance t-tests to check if the values for the reaction constants for TNBP and TSBP from the 3D fits can be determined to be different. For the 38 degrees of freedom of our samples and an alpha level of 5 %, the t-value for rejection is 2.024. For the kinetic parameters, the calculated t-value from the equal variance t-tests is:

	t-value
--	---------

$\ln(A)$	114.5
E_a	2.8e-08
ΔV^*	1.7e-07

This means we can confidently reject the hypothesis of pre-exponential factors being equal for TNBP and TSBP, while the values for E_a and ΔV^* are essentially identical.

Originally, we also had the question on the novelty of the current manuscript, compared to the previous paper of Zhang et al. ACS Appl. Mater. Interfaces 12, 6662 (2020). The authors pointed out two differences between the current manuscript and the published paper. One is that these two papers are studying different compounds with different decomposition mechanism. We leave it to the editors to decide whether this is a significant enough result to publish in this journal. The second claim to novelty is that this the first attempt of 3D fitting using Bell's model. As explained above, we think it is still not very convincing that 3D fitting is superior to 2D fitting or that such fit is even reasonable for the data obtained from current simulations.

The main novelty is of this work is that we have demonstrated the possibility to extract mechanochemical rate constants of antiwear additives using nonequilibrium molecular dynamics simulations. To this point, this has required difficult, time-consuming, and expensive experiments. The fact that we can differentiate between additives with different alkyl substituents using computer simulations is extremely valuable. Moreover, our results show good agreement with recent experiments using similar antiwear additives.

To our knowledge, the number of rate-temperature-stress data points (20 per molecule) is more than any previous experimental or simulation study of mechanochemistry processes. Thus, using 3D fitting to the Bell model, we are able to identify the cause of the different rates of two compounds that differ only in containing primary/secondary alkyl groups, i.e. the pre-exponential factor.

In the revised manuscript, we present both the 2D and 3D fits and the resulting parameters in full. The reviewer (and readers) can judge for themselves which is more useful. For our purposes, i.e. differentiating the mechanochemical behaviour of the two compounds, we feel that the 3D method is best-suited since it yields a significant difference between the compounds valid over the range of experimentally-relevant conditions studied here (see above).

Finally, we were also not convinced by the discussion of the transition state theory and collision theory presented in the original manuscript, as the authors did not provide supports

of the theory from simulations. In the revised version, the authors provided additional simulation results, i.e., Supplementary Figure 4., which show the number density profiles of the phosphate ester atoms between the surfaces. However, the interpretation of these plots is unclear. According to the authors, the peak of the number density profile for C in TNBP near the surface is split, whereas the number density profile of C in TSBP has a single peak near the surface. The authors write that the split peak of TNBP is an indication that C-O cleavage occurs only for some fraction of C atoms. The single peak of TSBP according to the authors implies that a large fraction of molecular collisions has led to C-O cleavage. However, as shown in Supplementary Fig. 4 (b), the single C peak of TSBP near the surface overlaps with O peak, while the split peak of TNBP in (b) does not overlap of O peak. This seems to support the opposite conclusion, i.e., there are more C-O cleavage for TNBP. Either way, the authors should justify better their conclusions from the density profiles presented in the revised manuscript.

The overlap in the number density profiles does not show what atoms are bonded together, but rather which are in located in the same atomic layer parallel to the sliding surfaces (in the z-direction). Whilst it is true more C and O overlap for TSBP than TNBP, Figure 2 in the manuscript clearly shows that more C-O bonds are broken for TSBP than TNBP. This point has been clarified in the revised manuscript.

We have added more information as to what physicochemical understanding can reasonably be expected to be extracted from the empirical modified Arrhenius (Bell) and Eyring model parameters fit to the data from the NEMD simulations. While some relevant quantities, e.g. then number of molecule-surface collisions at gas-solid interfaces, have been extracted previously (<https://doi.org/10.1103/PhysRevE.87.022119>), this is much more challenging (and to our knowledge has not been achieved) for liquid-solid interfaces. This is because collisions are more difficult to define from MD simulation of high-density systems. However, based on previous adsorption experiments of ZDDPs from base oil solution (<https://doi.org/10.1080/05698198708981753>), it is reasonable to assume that the collision rate is very similar for TNBP and TSBP.

Additional Information

In addition to the points raised by the reviewer, we performed additional MD simulations without sliding (Supplementary Figure 1) to compare the effect of normal stress and shear stress on reactivity. These simulations showed much lower reactivity without sliding, which confirms experimental suggestions that the shear stress, rather than the normal stress, controls the mechanochemical reactivity of alkyl phosphates.

REVIEWERS' COMMENTS:

Reviewer #2 (Remarks to the Author):

The authors have performed additional simulations and generally addressed the technical issues that we raised in our previous reviews. We are not convinced by the claims of novelty, but we leave it up to the editor to decide.